# Epilepsy-linked kinase CDKL5 phosphorylates voltage-gated calcium channel Cav2.3, altering inactivation kinetics and neuronal excitability

Marisol Sampedro-Castañeda [1,5] ✉, Lucas L. Baltussen [1,4,5], André T. Lopes [1], Yichen Qiu[2], Liina Sirvio [1], Simeon R. Mihaylov [1], Suzanne Claxton [1], Jill C. Richardson[3], Gabriele Lignani [2] & Sila K. Ultanir [1] ✉

Developmental and epileptic encephalopathies (DEEs) are a group of rare childhood disorders characterized by severe epilepsy and cognitive deficits. Numerous DEE genes have been discovered thanks to advances in genomic diagnosis, yet putative molecular links between these disorders are unknown. CDKL5 deficiency disorder (CDD, DEE2), one of the most common genetic epilepsies, is caused by loss-of-function mutations in the brain-enriched kinase CDKL5. To elucidate CDKL5 function, we looked for CDKL5 substrates using a SILAC-based phosphoproteomic screen. We identified the voltage-gated $Ca^{2+}$ channel Cav2.3 (encoded by *CACNA1E*) as a physiological target of CDKL5 in mice and humans. Recombinant channel electrophysiology and interdisciplinary characterization of Cav2.3 phosphomutant mice revealed that loss of Cav2.3 phosphorylation leads to channel gain-of-function via slower inactivation and enhanced cholinergic stimulation, resulting in increased neuronal excitability. Our results thus show that CDD is partly a channelopathy. The properties of unphosphorylated Cav2.3 closely resemble those described for *CACNA1E* gain-of-function mutations causing DEE69, a disorder sharing clinical features with CDD. We show that these two single-gene diseases are mechanistically related and could be ameliorated with Cav2.3 inhibitors.

Developmental and epileptic encephalopathies (DEE) are characterized by severe early-onset epileptic activity accompanied by global developmental and cognitive impairments. The genetic aetiology of approximately 90 DEEs has been identified[1], but targeted therapies remain scarce. Elucidating shared pathways in infantile-onset epilepsies will reveal molecular links between disease genes and greatly advance common targeted treatments.

Cyclin dependent kinase like-5 (CDKL5) is a brain-enriched serine-threonine kinase. De novo loss-of-function mutations in the X-linked *CDKL5* gene, including missense, nonsense and insertions/deletions,

[1]Kinases and Brain Development Lab, The Francis Crick Institute, 1 Midland Road, London NW1 1AT, UK. [2]Department of Clinical and Experimental Epilepsy, UCL Queen Square Institute of Neurology, Queen Square House, London WC1N 3BG, UK. [3]Neuroscience, MSD Research Laboratories, 120 Moorgate, London EC2M 6UR, UK. [4]Present address: Laboratory for the Research of Neurodegenerative Diseases (VIB-KU Leuven), Department of Neurosciences, ON5 Herestraat 49, 3000 Leuven, Belgium. [5]These authors contributed equally: Marisol Sampedro-Castañeda, Lucas L. Baltussen. ✉e-mail: marisol.sampedro-castaneda@crick.ac.uk; sila.ultanir@crick.ac.uk

cause CDKL5 deficiency disorder (CDD) (OMIM 300672, 300203)[2–5]. Pathogenic missense mutations, almost exclusively located in the kinase domain, indicate that kinase activity is critical for CDD pathology[6–9]. CDD is characterized by infantile-onset, intractable seizures, profound neurodevelopmental impairment in motor and sensory function, impaired language acquisition and autonomic disturbances[3,5]. The estimated incidence for CDD is 1/42,000 live births each year[10–12] with approximately 80% being female patients, making it one of the most common types of genetic childhood epilepsy[13,14].

CDKL5 is highly enriched throughout the forebrain with expression starting at late embryonic stages in rodents and humans[15–19]. Low levels of expression were reported in other tissues[16]. Phosphorylation targets include microtubule binding proteins EB2 and MAP1S[6,20] and transcriptional regulators ELOA[21] and SOX9[22], showing its diverse cellular roles. Cdkl5 knockout (KO) neurons have deficits in synaptic differentiation and connectivity, dendritic spine development, neurite outgrowth, cilia elongation and microtubule dynamics[23–30]. However, molecular mechanisms linking CDKL5 loss to neuronal hyperexcitability remain unknown. The identification of key physiological targets of CDKL5 directly involved in the regulation of cellular excitability may help elucidate epileptogenic mechanisms and lead to effective therapies.

Voltage-gated calcium channels (VGCC) have a key role in physiology, driving neuronal excitability and allowing influx of second messenger $Ca^{2+}$ ions in response to membrane depolarization[31–33]. VGCC subtype Cav2.3 is formed by the ion-conducting α1E subunit (encoded by CACNA1E) in complex with accessory subunits from the Cav β and α2δ families. Cav2.3 is highly expressed in the CNS, where it is localized at both pre- and post-synaptic sites in neurons, as shown functionally and by immuno-EM[34–36]. CACNA1E knockout mice demonstrated that Cav2.3 mediates neuronal R-type $Ca^{+2}$ currents[36–38], so-called due to their "resistance" to organic VGCC channel inhibitors[39,40]. Cav2.3 is dynamically regulated by phosphorylation at multiple sites[41,42]. In general, VGCC phosphorylation is an important mechanism by which neuromodulators and their G-protein coupled receptors (GPCRs) exert their effect on cellular excitability[43].

De novo heterozygous missense mutations in CACNA1E cause an infantile epileptic encephalopathy (OMIM 618285)[44,45]. Cav2.3 variants studied in vitro have altered gating and/or inactivation kinetics, leading to gain-of-function (GoF) of Cav2.3[44,45]. Patients with CACNA1E mutations have overlapping clinical phenotypes with CDD, such as intractable seizures, profound intellectual disability and hypotonia (Supplementary Table 1). Single-gene DEEs such as this one are estimated to impact approximately 1/2100 live births[10], but their real prevalence is unknown as more genes are being identified rapidly due to recent advances in genetic diagnosis[46,47]. Despite overlapping clinical features, potential molecular links between these rare disorders remain to be elucidated.

In this work, we apply for the first time a global phosphoproteomics approach in Cdkl5 KO mice and identify and validate Cav2.3 as a substrate of CDKL5 in human and mouse neurons. Through our analysis of Cav2.3 function and characterization of a novel Cav2.3 phosphomutant mouse line, in which CDKL5 phosphorylation site is mutated, we reveal that lack of phosphorylation leads to Cav2.3 gain-of-function and hyperexcitability. Our results indicate that developmental and epileptic encephalopathies caused by CDKL5 and CACNA1E are related at the molecular level. We propose Cav2.3 as a converging therapeutic target for these disorders.

## Results

### Cav2.3 is a CDKL5 phosphorylation substrate in mice and humans
We used stable isotope labelling of amino acids in cell culture (SILAC)[48] to differentially label newly synthesized proteins in primary cortical cultures from wild type (WT) and Cdkl5 knockout (KO) mice[49] and compare changes in protein phosphorylation levels (Fig. 1a). As expected in Cdkl5 KO neurons, we observe a significant reduction in CDKL5 pS407 and in EB2 pS222, a known CDKL5 phosphorylation target[20]. Phosphorylation of EB2 S221 was also decreased, indicating co-regulation of these previously reported neighbouring sites. More importantly, a potential new target protein, the ion channel Cav2.3, showed strongly reduced phosphorylation at S15 (pS15), in Cdkl5 KO neurons. Strikingly, the murine Cav2.3 S15 site (corresponding to S14 in humans) matches the CDKL5 RPXS/T* consensus motif[6,20], suggesting direct phosphorylation (Fig. 1a, b). Serine 15, located in the N-terminus of the α1E channel pore subunit (Fig. 1b), is conserved in humans (S14) but absent in the related Cav2.1 (CACNA1A) and Cav2.2 (CACNA1B) channels (Fig. 1b).

To test if CDKL5 can phosphorylate Cav2.3 S15/S14, we generated a phosphospecific antibody targeting this site using the mouse sequence. Overexpression of Cav2.3 (& ancillary subunits) together with the CDKL5 kinase domain in HEK293 cells shows strong Cav2.3 phosphorylation in vitro (Cav2.3 pS14, Fig. 1c, f). This phosphorylation is absent when overexpressing phosphomutant Cav2.3 or kinase-dead CDKL5 mutant, demonstrating the specificity of the antibody and the dependence of phosphorylation at this site on CDKL5 kinase activity. Importantly, full-length CDKL5 also phosphorylated Cav2.3 in vitro (Supplementary Fig. 1a, b). Cav2.3 phosphorylation is consistently reduced in the cortex of Cdkl5 KO mice at postnatal day P20 (Cav2.3 pS15), validating the substrate in vivo (Fig. 1d, g). Finally, Cav2.3 pS14 was decreased in human iPSC-derived neurons from CDD patients (Fig. 1e, h), demonstrating altered phospho-regulation of this target in CDD. Together, these results show that Cav2.3 is a bona fide CDKL5 phosphorylation target in mice and humans.

### Loss of Cav2.3 Ser14 phosphorylation slows baseline channel kinetics and amplifies muscarinic regulation
Cav2.3 is causally linked to epilepsy by two main lines of evidence. First, channel deletion renders mice resistant to tonic-clonic seizures[50–52]. On the other hand, Cav2.3 GoF leads to severe epileptic encephalopathy in humans[44,45]. The overlap in clinical manifestations between CACNA1E and CDD patients, including seizures and neurodevelopmental delays, raises the possibility that Cav2.3 regulation by CDKL5 might affect channel function and in turn neuronal excitability.

To test the effect of pS14 on Cav2.3 channel properties we used a HEK293 cell line stably expressing the human auxiliary subunits β3 and α2δ1 and transiently transfected human α1E and full-length CDKL5 plasmids. Whole-cell patch-clamp recordings of Cav2.3 $Ba^{2+}$ currents revealed slower decay kinetics (inactivation τ) in S14A mutant channels (Fig. 2a–c). Notably, in absence of CDKL5, WT Cav2.3 decay kinetics were indistinguishable from S14A Cav2.3 mutant (Fig. 2d). We also found a statistically significant increase in whole-cell current density at maximum activation voltages in S14A when compared to WT Cav2.3 in presence of CDKL5 (Fig. 2e). To account for differences in biophysical properties that may be introduced by α1E association with a different β subunit[53,54], we conducted parallel experiments in a HEK293 cell line stably expressing human α2δ1 and β1b, another Cavβ subunit associated with α1E in the brain[55]. We used $Ca^{2+}$ as charge carrier to examine channel properties in a more physiological context. Decay kinetics of WT Cav2.3 currents with β1b were substantially slower than currents with β3 (Supplementary Fig. 2a, b vs. Fig. 1a, b), as previously reported[53]. In absence of CDKL5 phosphorylation (S14A α1E mutant or kinase-dead CDKL5 conditions), Cav2.3 current inactivation was further slowed (Supplementary Fig. 2a–c), recapitulating our results with β3 (Fig. 2a–c). Current density at maximal activation voltages did not reach significance in the β1-expressing cells, where currents were generally larger (Supplementary Fig. 2d). Finally, we observed no changes in half maximal activation and inactivation potential or time

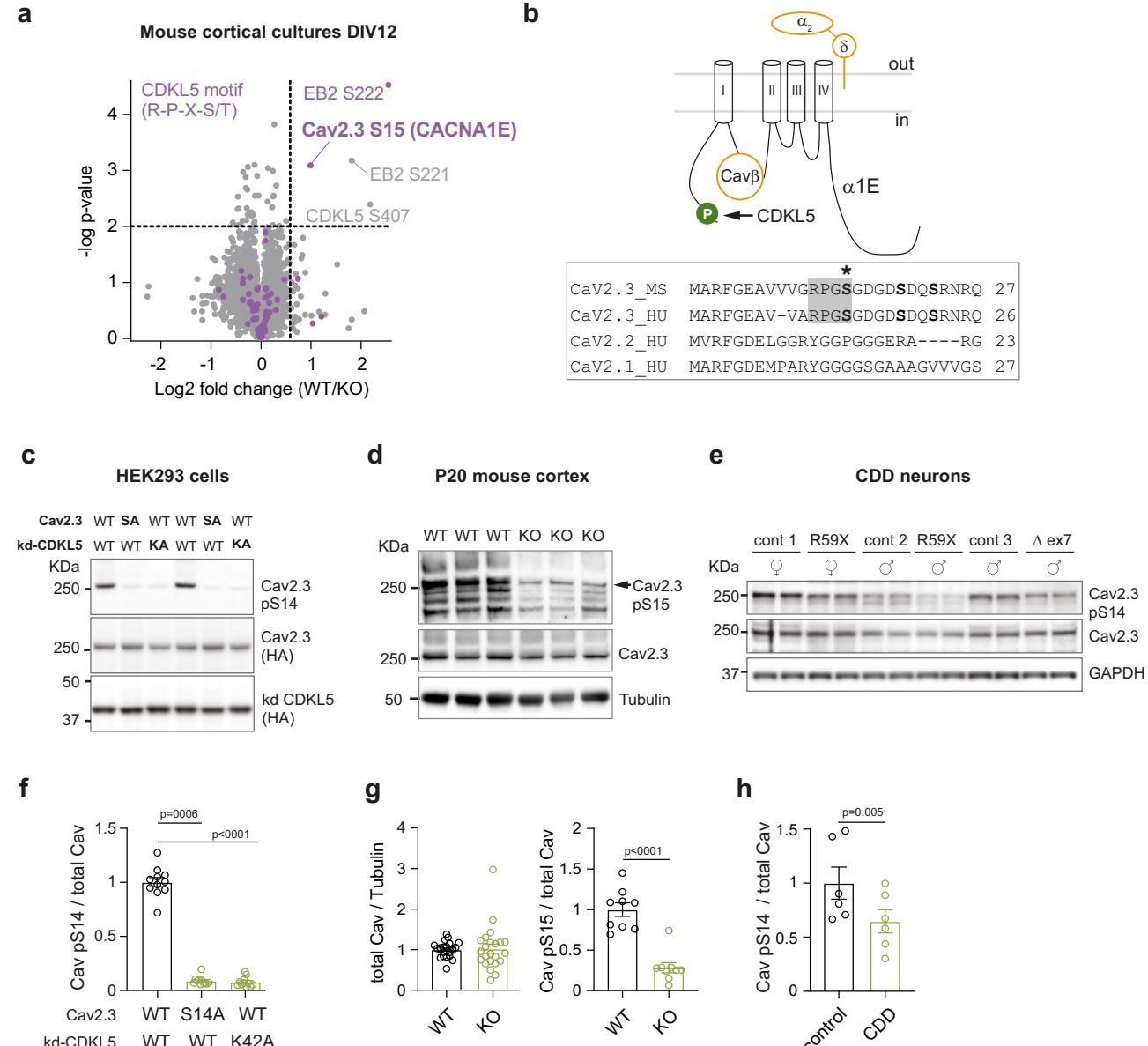

**Fig. 1 | Identification and validation of Cav2.3 as substrate of CDKL5 kinase.**
**a** Volcano plot of differential phosphoprotein levels between WT and *Cdkl5* KO primary neurons, obtained using SILAC based quantitative proteomics (3 embryos/genotype). Each point represents one peptide; those with CDKL5 consensus motif RPXS/T are highlighted in magenta. The *x* axis shows log2 transformed fold change in phosphopeptide levels between genotypes and the *y* axis shows significance as -log10 transformed *p* value (one sample *t* test). Significance was established as indicated by the dotted lines ($p < 0.01$, fold change 1.5, one sample *t* test). **b** Top: Schematic depiction of Cav2.3 formed of α1E channel pore subunit and associated proteins Cavβ and α2δ. CDKL5 phosphorylation site at α1E S15/14 is highlighted in green. Bottom: Alignment of proximal N-terminus of mouse and human Cav2.3 and related human Cav2.1 and 2.2 channel proteins. *S is the site of CDKL5 phosphorylation. **c** Western blot of HEK293 cells stably expressing human β3/α2δ1 subunits and transiently co-transfected with human HA-α1E (Cav2.3: WT or S14A mutant) and HA-CDKL5 kinase domain (kd CDKL5: WT or K42A mutant). S14 phosphorylation (pS14 Cav2.3) was detected with a custom made phosphoantibody. Example blots

derive from two gels run and processed in parallel. **d** Western blot of P20 cortices from WT and *Cdkl5* KO mice. Example blots derive from three gels: phospho and total Cav were run and processed in parallel; control α-tubulin was obtained from a separate experiment using the same samples and loading volume. **e** Western blot of iPSC-derived forebrain neurons from three CDD patients and related controls (parents) at 6 weeks of differentiation. Gender and *CDKL5* mutation are specified[90]. Example blots derive from two gels run and processed in parallel. **f** Quantification for immunoblot in (**c**) ($p < 0.0001$ and $p = 0.0006$, Kruskal-Wallis ANOVA & Dunn's test, $n = 11$ for each condition: 5 transfections, 2–3 technical replicates).
**g** Quantification for immunoblot in (**d**) (total Cav: $p = 0.25$, $n = 23$, 6 blots, 6 mice/genotype; pCav: $p < 0.0001$, two tailed unpaired *t* test, $n = 9$, 3 blots, 6 mice/genotype; no technical replicates). **h** Quantification for WB in (**e**) ($p = 0.0051$, two tailed paired *t* test, $n = 6$, 3 control/patient pairs, 4 technical replicates). For antibodies used see Methods. Data is presented as mean ± S.E.M. All source data is provided in a Source Data File.

course of recovery from fast inactivation in β3- or β1-expressing cells (Fig. 2f and Supplementary Fig. 2e, f).

Altogether, our results indicate that CDKL5-mediated Ser14 phosphorylation of α1E speeds up the inactivation of human Cav2.3 and suppresses current amplitude, independent of Cavβ subtype or charge carrier. Consequently, absence of CDKL5 leads

to Cav2.3 GoF, with larger and more prolonged currents. A subset of *CACNA1E* GoF mutations described in human DEE patients also cause slower inactivation kinetics and/or increased current density[44], mirroring the phenotypes observed in the unphosphorylated channel. Thus, the functional deficits in Cav2.3 observed in absence of CDKL5 phosphorylation match those

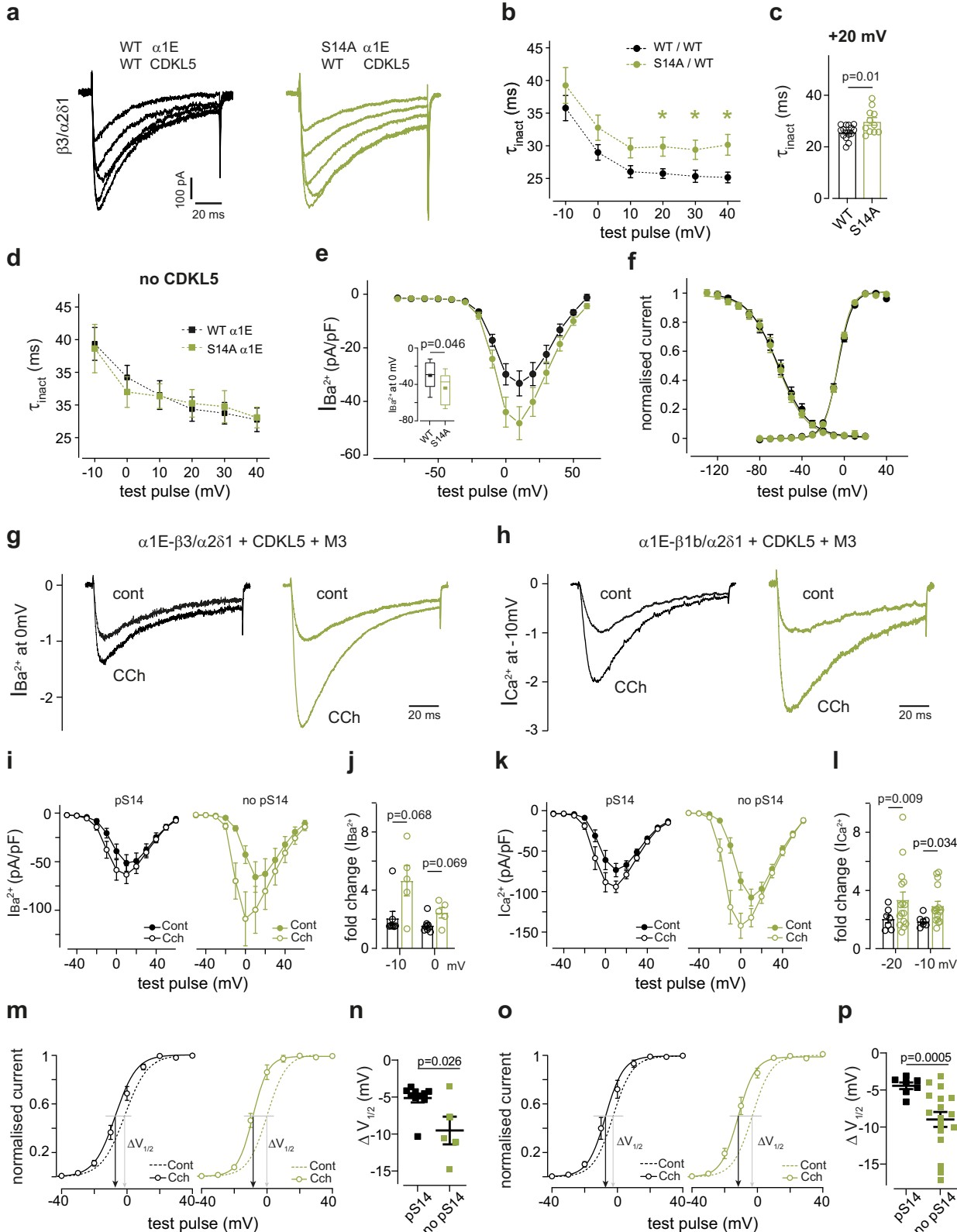

caused by pathological point mutations in Cav2.3 and could partly explain the pathophysiology of CDD.

Cav2.3 channels are downstream targets of muscarinic acetylcholine receptors, which modify channel function predominantly via PKC-mediated phosphorylation, leading to current enhancement in HEK293 cells and neurons[56–58]. Given the observed GoF in S14A Cav2.3, we hypothesized that muscarinic receptor enhancement may also be

affected by CDKL5 phosphorylation. To test this, we co-expressed Cav2.3, CDKL5 and muscarinic acetylcholine receptor type 3 or 1 (M3 or M1), which are prominent brain subtypes known to regulate neuronal Cav2.3 post-synaptically, as well as GFP for visualization. We confirmed that 4 plasmids can be co-transfected with our protocol by immunostaining for each protein (Supplementary Fig. 3a, b). As expected, application of the non-hydrolysable cholinergic agonist

**Fig. 2 | Functional characterization of phospho-Ser14 Cav2.3 in HEK 293 cells.**
**a** Depolarization-evoked current responses in HEK293 cells stably expressing human β3/α2δ1 subunits and co-transfected with human Cav2.3 (WT α1E or S14A α1E) and FLAG-CDKL5 full length (WT CDKL5). Colours denote different construct combinations. Traces show steps from −10 to +30 mV from −80 mV; Ba$^{2+}$ was charge carrier (**b**) Open channel inactivation tau (τ$_{inact}$) for Cav2.3 with (WT/WT, $n = 13$–15 recordings from −10 to +40) and without (S14A/WT, $n = 11$–13 recordings from −10 to +40) CDKL5 phosphorylation (*$p = 0.02$, 0.03 & 0.01 at +10, +20 & +40 mV Two-Way ANOVA, Fisher's LSD). Some voltage steps were excluded as described in Methods. Data were acquired using 100 ms voltage steps from −80 mV in +10 mV increments every 10 s. **c** Individual data points for τ$_{inact}$ at +20 mV (WT/WT $n = 14$, S14A/WT $n = 11$; two-tailed unpaired $t$ test). **d** WT or phophomutant S14A Cav2.3 inactivation in the same cell line in absence of CDKL5 (both conditions $n = 7$–9; $p > 0.05$ Two-Way ANOVA); Ca$^{2+}$ was charge carrier. **e** Current voltage relationship for cells in (**b**). Inset: comparison of current density at 0 mV (inset, two-tailed unpaired $t$ test). **f** Normalised Cav2.3 current conductance and voltage dependence of inactivation for the same transfection conditions. Activation V$_{1/2}$, n: WT/WT −6 ± 1 mV, 15; S14A/WT −6 ± 1 mV,12; Inactivation V$_{1/2}$, n: WT/WT

−61 ± 3 mV,10; S14A/WT −62 ± 2 mV,11 ($p > 0.05$, two tailed unpaired $t$ tests). For inactivation protocol see Methods. Solid lines are Boltzman fits to the average data points (**g**) Normalised Ba$^{2+}$ currents in the β3/α2δ1-cell line or (**h**) Ca$^{2+}$ currents in the β1/α2δ1-cell line, co-transfected with α1E (WT or S14A), WT CDKL5 and muscarinic type 3 receptor (M3), in presence and absence of carbachol (CCh) 30 and 10 µM, respectively. HP = −100 mV. **i** IV curves and (**j**) CCh-induced change in current amplitude for experiment in (**g**) (pS14 $n = 10$, no pS14 $n = 5$, Two-Way ANOVA, Fisher's LSD). **k** IV curves and (**l**) CCh-induced change in current amplitude for experiment in (**h**) (pS14 $n = 7$, no pS14 $n = 16$, Two-Way ANOVA, Fisher's LSD). **m** Conductance plots for experiment in (**g**) and **n** Corresponding change in activation (V$_{1/2}$: −5 ± 0.6 mV ($n = 10$) vs. −10 ± 1.9 mV ($n = 5$), two-tailed Kolmogorov-Smirnov). **o** Conductance plots for experiment in (**h**) and **p** corresponding change in activation (V$_{1/2}$: −4 ± 0.5 mV ($n = 7$) vs −9 ± 1 mV ($n = 16$), two-tailed Welch's $t$ test). From (**l**–**p**), 'pS14' refers to WT α1E/WT CDKL5; 'no pS14' refers to S14A α1E /WT CDKL5 and α1E WT/K42R* CDKL5 conditions pulled together. Data is presented as mean ± S.E.M or in box plots representing minimum, maximum, median, 25/75 percentile and mean (indicated by a marker). Source data are provided as a Source Data file.

carbachol substantially increased current amplitude regardless of Ser14 phosphorylation in both β3 and β1 stable cell lines (Fig. 2g–l). The current increase with carbachol was 1.5- to 2-fold greater in cells without phospho-Ser14 at −10 mV (Fig. 2g, h, j, l), reflecting Cav2.3 gain-of-function in absence of phosphorylation. The half-maximal activation potential (V$_{1/2}$) was shifted in the hyperpolarizing direction by 5 mV for WT channels in both cell lines, a described effect of muscarinic receptor activation[57,59]. Interestingly, without Ser14 phosphorylation this shift was significantly greater, implying an effect of phospho-Ser14 on the modulation of Cav2.3 gating (Fig. 2m–p). This represents an additional gain-of-function due to loss of CDKL5, a phenotype comparable to pathogenic *CACNA1E* mutations that alter V$_{1/2}$[44].

Upon GPCR activation, Cav2.3 channels can be suppressed by G$_{βγ}$ subunits. This effect has been observed in Gαq-coupled receptors including M1/M3 in HEK293 cells[57] and CA1 neurons[56]. The Cav2.3 N-terminus and Cavβ are involved in G$_{βγ}$ regulation[60,61]. For this reason, we investigated the possibility that the enhanced carbachol effects in absence of N-terminal pS14 could be due to reduced G$_{βγ}$ suppression. To distinguish between PKC-mediated enhancement and G$_{βγ}$ suppression, we boosted PKC activity prior to carbachol application using phorbol 12-myristate 13-acetate (PMA), reliably increasing Cav2.3 currents[57,62]. With further addition of carbachol we found no evidence of muscarinic inhibition of human WT Cav2.3 channels (Supplementary Fig. 2g, h). Moreover, co-expression of Cav2.3 and Gi-coupled dopamine D2 receptors (Supplementary Fig. 3c) to further probe G$_{βγ}$ inhibition in our system, revealed significant reduction in peak currents upon D2 activation with quinpirole[60]. However, there was no difference in inhibitory modulation with or without Ser14 phosphorylation (Supplementary Fig. 2i, j). The above manipulations suggest that G$_{βγ}$ is not involved in pS14-dependent Cav2.3 GoF.

Our results describe two Cav2.3 gain-of-function (GoF) mechanisms in the absence of CDKL5 phosphorylation: slower decay kinetics and a greater hyperpolarizing shift in voltage-dependence of activation (V$_{1/2}$) upon muscarinic activation, both reflected in increased Ca$^{2+}$ influx through Cav2.3 channels. These GoF features are reminiscent of those described for pathological Cav2.3 variants.

### Altered R-type current inactivation and increased excitability in neurons from Cav2.3 S15A phosphomutant mice

To investigate the role of Cav2.3 S15 phosphorylation in neurons, we generated Cav2.3 S15A phosphomutant mice using CRISPR-Cas9 genome editing. We found that the total S15 phosphorylation levels detected by our phosphoantibody were reduced to background levels of <10% of control levels in homozygous S15A mice (HOM S15A) (Fig. 3a). Heterozygous S15A (HET S15A) mouse brains had approximately 50% of S15 phosphorylation, as expected, while total Cav2.3 levels were not changed (Fig. 3a–c). We compared pS15

phosphorylation in *Cdkl5* KO mice and phosphomutant mice in parallel on the same experiment and found that 15–20% of phosphorylation remained in *Cdkl5* KO mice, potentially due to other kinases phosphorylating the same site (Supplementary Fig. 4a, b). To study the role of pSer15 we used HOM S15A mice.

Cav2.3 is highly expressed in the somato-dendritic region of CA1 hippocampal neurons[35] where it underlies most of the R-type current and regulates intrinsic excitability[36,56]. We performed whole-cell somatic recordings in acute slices from young mice and recorded pharmacologically isolated Cav2.3 Ca$^{2+}$ currents in these cells. When compared to wild type littermates (WT), HOM S15A mice showed significantly slower inactivation τ at maximal conductance voltages, recapitulating HEK293 cell results (Fig. 3d, e). Current amplitudes at the soma did not differ (Fig. 3f).

We next tested the effect of Cav2.3 S15A on neuronal excitability and cholinergic neuromodulation in response to depolarising current injections in adult CA1 pyramidal cells. Resting membrane potential (V$_{rest}$), input resistance (R$_{in}$) and firing properties were identical between WT and S15A mice under baseline conditions (Fig. 3g–i). Following bath application of carbachol, we observed a 4 mV depolarisation in V$_{rest}$ and a 25% increase in R$_{in}$ (Fig. 3g), changes that were comparable between WT and S15A neurons (Supplementary Fig. 5a). These alterations are likely due to conductances other than Cav2.3, in line with previous reports[59,63,64].

In neurons, cholinergic activation can elicit sustained depolarizing plateau potentials (DPPs): all-or-none, large amplitude depolarizations that outlast stimulus-evoked firing[59,65,66]. A muscarinic medium duration afterdepolarization (ADP) is also reported in CA1 pyramidal cells[67]. Up-regulation of Cav2.3-mediated R-type currents is required for DPPs and ADPs in hippocampus[68] and cortex[66]. We inspected these Cav2.3-dependent hyper-excitability measures in S15A mice. Carbachol application significantly increased action potential firing frequency in WT and S15A neurons (Fig. 3h–k). At current injections greater than 250 pA, however, phosphomutant neurons responded with fewer spikes and depolarization block towards the end of the stimulus step, indicating greater underlying depolarization. The percentage of cells exhibiting at least one DPP or all-or-none afterdepolarization (ADP) > 8 mV upon carbachol treatment was significantly higher in S15A mice, showing an enhanced depolarization response (Fig. 3j, k). Importantly, the occurrence of DPP/large ADPs was shifted towards lower current injections in S15A cells (Fig. 3l). Greater perisomatic depolarization in S15A neurons was also reflected in a larger ADP at 100 ms post stimulus and more spike attenuation (Supplementary Fig. 5b, c). Together with the enhanced carbachol-mediated hyperpolarizing shift in V$_{1/2}$ observed in phosphomutant channels in HEK293 cells (Fig. 2j, l), these increased neuronal depolarizations in Cav2.3 phosphomutant mice suggest

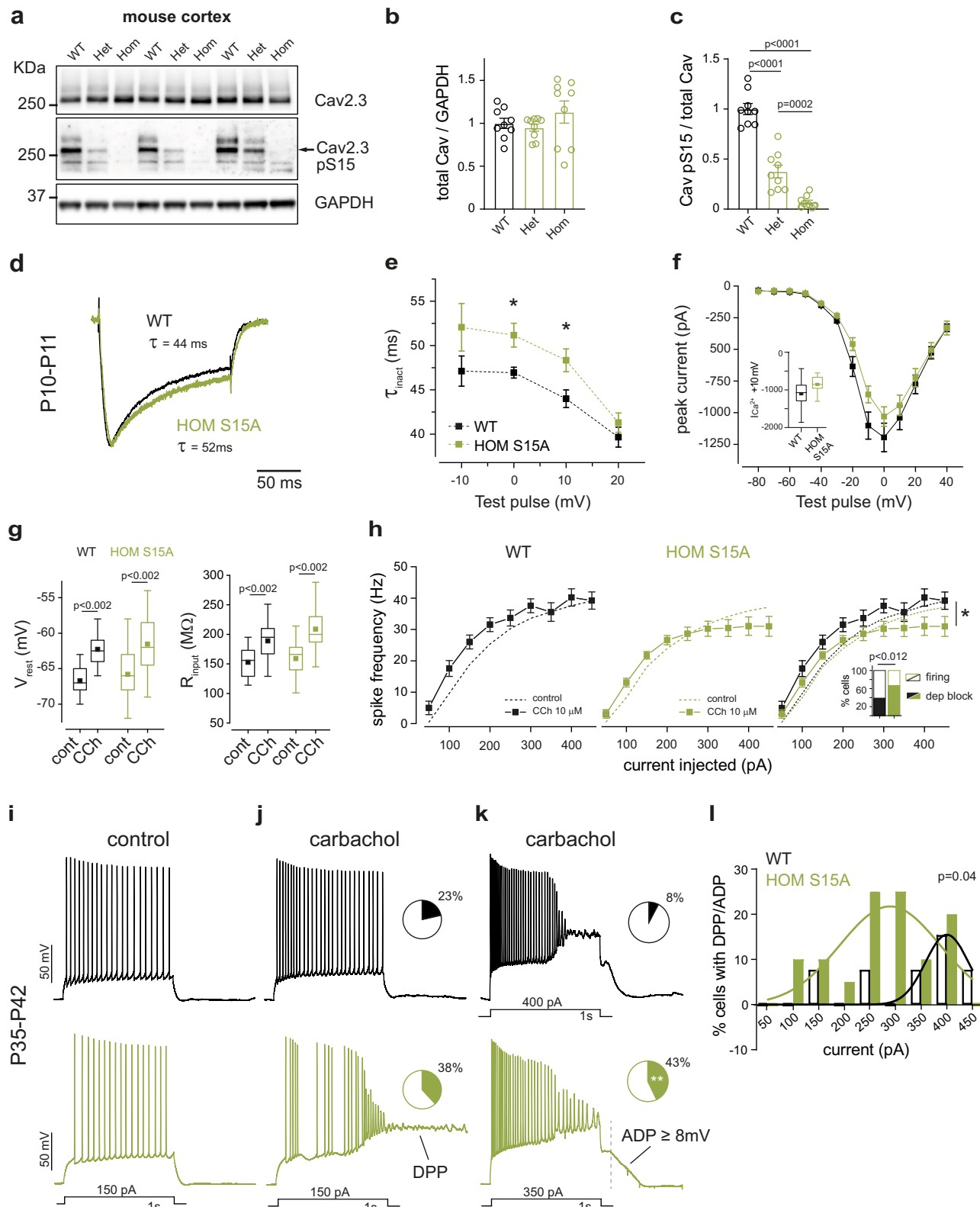

facilitation of gating of endogenous Cav2.3 without S15 phosphorylation upon cholinergic agonist application.

We tested if other conductances known to be regulated by carbachol are altered in Cav2.3 phosphomutants. The medium- and slow-afterhyperpolarizations (AHPs) are critical for spike frequency regulation and their underlying $K^+$ and mixed-cationic currents are targets of the muscarinic system[63,69,70]. Baseline AHPs elicited by a short spike

burst were not different between WT and S15A mice and were equally suppressed by carbachol application (Supplementary Fig. 5d). We also analysed baseline small-conductance $Ca^{2+}$-activated $K^+$ (SK) currents and spontaneous excitatory synaptic currents (EPSCs), known to be functionally coupled to[71] or modulated by[72] Cav2.3 in CA1 neurons. We hypothesized that these may be affected given the prolonged Cav2.3 currents in S15A mice, but our measurements show no differences

**Fig. 3 | Physiological properties and cholinergic neuromodulation of hippocampal CA1 neurons from WT and phosphomutant mice. a** WB validation of CRISPR-generated Cav2.3 Ser15Ala mutant mice using a pS15 phosphoantibody in cortical lysates from 5–6-week-old WT, Heterozygous S15A and Homozygous S15A mice. Example blots derive from two gels run and processed in parallel. **b** Quantification of total Cav2.3 ($p > 0.05$ One-Way ANOVA) and **c** Relative phospho-S15 Cav2.3 expression (One-Way ANOVA, Fisher's LSD; $n = 9$, 3 mice/genotype, 3 technical replicates). **d** Normalised R-type $Ca^{2+}$ current in neurons from young mice of each genotype during a step from −60 to 0 mV in presence of $Na^+$, $K^+$, $Ca^{2+}$ and synaptic channel inhibitors. **e** R-type current inactivation tau ($\tau_{inact}$) at maximal activation voltages (*$p = 0.01$ Two-Way Repeated Measures ANOVA, Fisher's LSD) and **f** Average IV curve for all cells recorded ($p > 0.05$), ($n = 13$ neurons for both genotypes, 4–5 mice/genotype). Data for −10 mV step is shown in the inset ($p = 0.06$, two-tailed unpaired $t$ test). **g** Membrane resting potential ($V_{rest}$, left) and input resistance ($R_{input}$, right) of adult neurons ($V_{rest}$: WT = 14, HOM S15A $n = 20$; $R_{input}$: WT $n = 13$, HOM S15A $n = 17$; 7 mice/genotype) in control conditions and after 10 μM carbachol (CCh) application (two-tailed paired $t$ test). **h** Input/output curves for the same neurons (WT $n = 9$, HOM $n = 13$, *$p < 0.028$ Two-Way Repeated

Measures ANOVA) and percentage of cells with CCh-induced depolarization block (inset, WT $n = 13$, HOM $n = 21$, two-tailed binomial test). **i** Representative control action potential trains evoked in WT (top) and HOM S15A (bottom) CA1 cells by 1s-long current injections. Membrane was held at −65 mV. **j** Same cells as (**i**) upon application of CCh and representative depolarising plateau potential (DPP), observed at some current injections in both genotypes; insets: fraction of cells with at least one DPP in each genotype group (WT $n = 13$, HOM $n = 21$, $p > 0.05$, two-tailed binomial test). **k** Same cells as (**j**) at higher stimulation illustrating sustained depolarization and AP block (top) or attenuation (bottom) towards the end of the stimulus and long-lasting large amplitude afterdepolarizations (ADP). Like DPPs, these ADPs were present at some current injections in a fraction of cells in both groups of mice (inset, WT $n = 13$, HOM $n = 21$, **$p < 0.004$, two-tailed binomial test). **l** Percentage of cells displaying sustained DPPs or ADPs upon stimulus termination at each current injection step. Non-linear fits (least squares regression) were compared using the extra sum-of-squares F test ($p = 0.04$ indicates the data cannot be adequately fit with a single Gaussian). Data is presented as mean ± S.E.M or in box plots representing minimum, maximum, median, 25/75 percentile and mean is indicated by a marker. Source data are provided as a Source Data file.

---

between genotypes (Supplementary Fig. 5e–h). Therefore, we suggest that the increased carbachol-induced excitability of CA1 neurons in Cav2.3 S15A knock-in mice is specifically due to Cav2.3 GoF.

## Cav2.3 S15A phosphomutant mice have behavioural and EEG deficits

Multiple *Cdkl5* KO mouse models have been generated by deleting exons 4, 2 or 6, all of which led to loss of CDKL5 protein[15,18,49]. Numerous behavioural deficits were reported in *Cdkl5* deficient mice[15,49,73–80]. Thus, we investigated motor, social and cognitive behaviours in Cav2.3 HOM S15A phosphomutants to test in vivo consequences of loss of pS15 and compare with reported *Cdkl5* phenotypes. We noted sex and/or genotype specific effects in some behavioural assays, as detailed in Fig. 4 and Supplementary Fig 6. When monitored in a home-cage environment for prolonged periods, S15A male mice showed reduced locomotion and voluntary wheel use (Fig. 4a and Supplementary Fig. 6a, b), also observed in *Cdkl5* deficient mice under similar conditions, although in this case both males and females were affected[49]. In contrast, we do not observe overactivity or anxiety in the open field test (Supplementary Fig. 6c) as reported in *Cdkl5* models of both sexes[73–75,77,79]. Also, unlike *Cdkl5* deficient mice[15,73,75,79,80], S15A phosphomutants did not present hindlimb clasping (mean scores: males 0; females WT 0.041, HOM S15A 0, $p = 0.35$, $n = 8$ and 6, Welch's t test) and performed equally in the rotarod test (Fig. 4b) when compared to WT littermates. On the other hand, S15A female mice had impairments in sociability (Fig. 4c) which is altered in *Cdkl5* KO males[15,73,74,79] but not tested in females. Cognitive abilities in the Y-maze were unchanged in phosphomutants (Supplementary Fig. 6d) but we found deficiencies in memory formation and retention for S15A males in the Barnes maze task and for both phosphomutant sexes in the fear conditioning test (Fig. 4d–f and Supplementary Fig. 6e, f), indicating a clear overlap with *Cdkl5* models, where alterations in spatial and fear memory have been consistently reported for males[15,73,74,78,79] and females[77]. S15A male deficits in the Barnes maze are not related to locomotion (Supplementary Fig. 6f).

Given the high epilepsy incidence in human CDKL5 and CACNA1E DEE patients, we investigated potential seizures. We did not observe any spontaneous behavioural seizures in Cav2.3 S15A mice up to 40 weeks of age, as reported for adult *Cdkl5* KO mice[15,18,49,81] and rats[28], potentially reflecting species differences between rodents and humans. Curiously, seizures are observed in excitatory neuron-specific *Cdkl5* KOs or in older heterozygous females[81–83]. Furthermore, electrocorticogram (EcoG) activity recorded in a separate cohort of phosphomutants over two weeks using wireless transmitters[84] indicate no changes or sex specific effects in baseline ECoG (Fig. 5a). Following these control recordings, seizure susceptibility was analysed with

repeated low-dose kainic-acid (KA) injections[85] (Fig. 5b, c). We found no differences in maximal KA-induced seizure severity in S15A mice (Fig. 5b), similar to KA-induced seizures in *Cdkl5* KO male mice[18,49]. Interestingly, however, female S15A mice had a reduced threshold to stage 5 behavioural seizures along with an increase in ECoG activity during KA (Fig. 5c). We could not recover transmitter activity during KA for some male mice.

Finally, since heterozygous *Cdkl5* KO females have spontaneous seizures starting at approximately 20 weeks and increasing with age[81,83], we investigated the levels of Cav2.3 phosphorylation in brain by comparing 8- and 22-week-old mice. We found that Cav2.3 total levels are increased and relative levels of pS15 are reduced in 22-week-old mice (Supplementary Fig. 7), raising the possibility that increased Cav2.3 expression could contribute to late-onset epilepsy in heterozygous *Cdkl5* KO females.

In summary, our in vivo analyses of Cav2.3 S15A mice demonstrate similarities with CDD mouse models and patients, including social, motor and cognitive impairments, and to some extent increased seizure susceptibility. Taken together, these observations suggest that loss of pS15 contributes to the phenotypic features of CDD.

## Discussion

There are currently no disease-targeting therapies for CDD and the causes of aberrant excitability in absence of CDKL5 are unknown. Here we identify the alpha subunit of Cav2.3 channel complex as a CDKL5 phosphorylation substrate in mouse and human neurons. *CACNA1E* mutations cause severe early onset epilepsy in humans and there is some overlap between CDD and CACNA1E encephalopathy (Supplementary Table 1). These include a high incidence of intractable seizures, global developmental delay, intellectual disability, autistic behaviours, a range of motor phenotypes such as hypotonia, hyperkinetic movements and stereotypies, as well as sleep, visual and sensory disturbances. The cellular and circuit basis of these clinical manifestations remains to be dissected in both neurodevelopmental disorders. CDKL5 deficiency is multifactorial as many kinase targets exhibit reduced phosphorylation. Nevertheless, a consistent feature of the Cav2.3 variants associated with epilepsy studied to date is GoF of the channel, by shifting the activation voltage to more hyperpolarized potentials, altering maximal current and/or by slowing inactivation[44]. Our data shows that deletion of *Cdkl5* in mice results in a significant reduction in Cav2.3 S15 phosphorylation, which leads to GoF due to slowed inactivation and amplified gating modulation, similar to the effects observed in Cav2.3 variants. Thus, our results establish Cav2.3 overactivity as a common feature of CACNA1E (DEE69) and CDKL5 (DEE2) epileptic encephalopathies, potentially explaining some of the overlapping clinical features of these diseases.

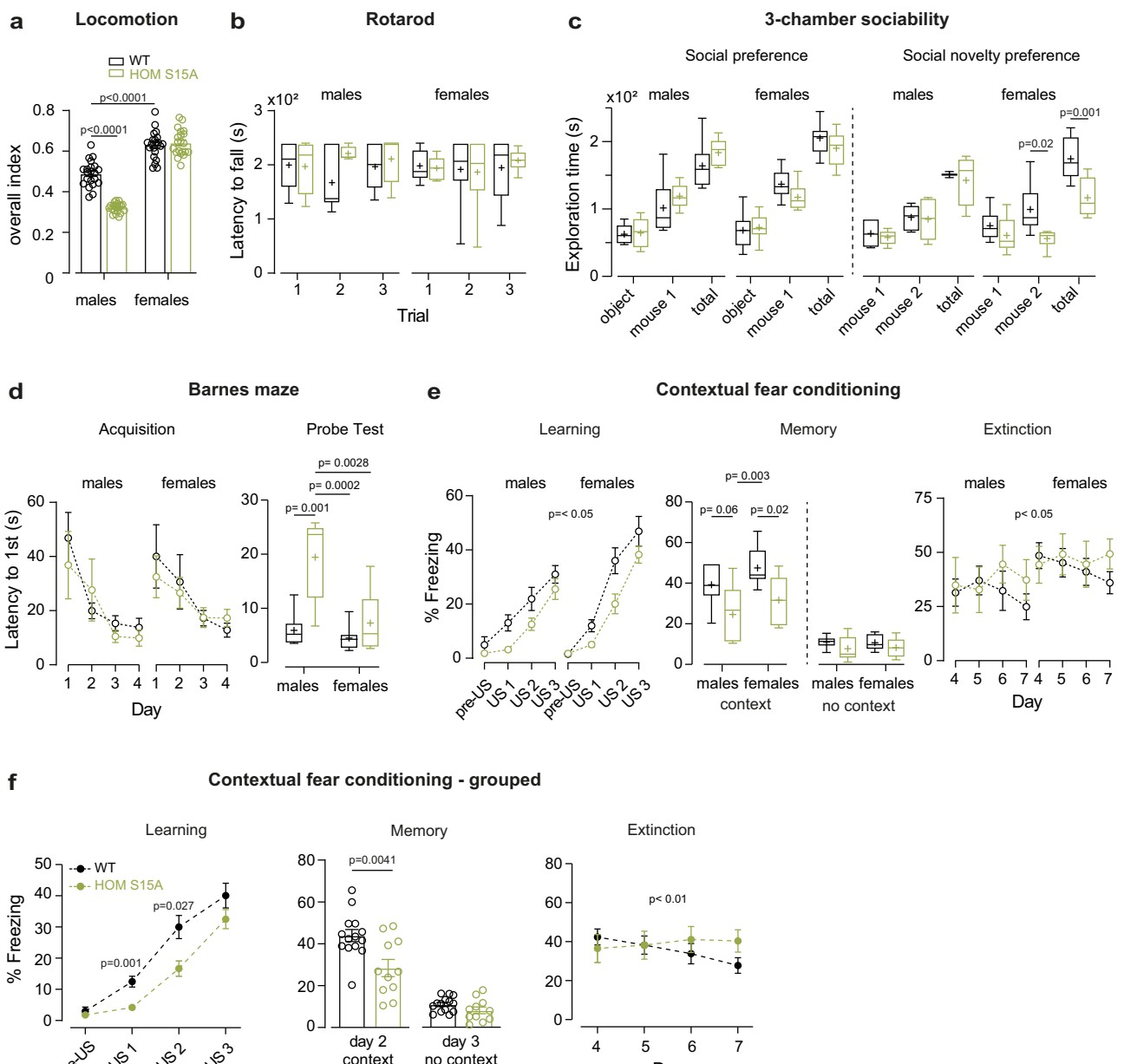

**Fig. 4 | Behavioural characterization of Cav2.3 S15A mice. a** Overall home cage night-time locomotion index of WT and HOM S15A mice over a three-week period (males WT $n = 5$, HOM $n = 4$; females WT $n = 4$, HOM $n = 5$; Two-Way ANOVA, Sidak's; sex x genotype $p < 0.0001$). **b** Accelerating rotarod performance for a separate cohort ($p > 0.05$, Two-Way ANOVA, Tukey's). Animal weights were equal: WT vs HOM [grams (n)], males 28.5 (6) vs 29.3 (5), females 21.8 (8) vs 21.3 (6); $p = 0.8$ & $p = 0.4$, respectively, two-tailed unpaired $t$ test. **c** Quantification of the social, non-social and total exploration times during a 10 min three-chamber sociability test in 2-phases for the same cohort as (**b**) (Two-Way ANOVA, Sidak's); social novelty test: sex x genotype $p = 0.04$, Three-Way ANOVA. In habituation trials, chamber occupancy was equal between groups. **d** Left: acquisition phase of the Barnes maze test with improved performance for both groups in the first 4 training days (same cohort as (**b**), $p > 0.05$, Two-Way ANOVA, Sidak's); right: memory assessment on probe day 5 (Two-Way ANOVA, Tukey's); sex x genotype $p = 0.01$, Three-Way ANOVA. **e** Freezing behaviour during learning day 1 (left), associative memory test day 2 + 3 (middle) and memory extinction test days 4–7 (right), in a fear-conditioning experiment with tactile and olfactory cues; Same cohort as (**b**): Day 1: males US1 $p = 0.08$, females US1 $p = 0.06$, US2 $p = 0.08$, Two-Way ANOVA, Tukey's; sex $p = 0.03$, genotype $p = 0.009$, Three-Way ANOVA. Day 2 + 3: Two-Way ANOVA, Fisher's LSD; for day 2 genotype $p = 0.005$ Three-Way ANOVA. Day 4–7: day x genotype $p = 0.04$ Three-Way ANOVA. **f** Grouped data for both sexes in the fear conditioning test. Same cohort as (**b**): Day 1: genotype x US $p = 0.04$ Two-Way ANOVA, Sidak's. Day 2 + 3: two tailed $t$-tests. Day 4–7: genotype x day $p = 0.0006$ Two-Way ANOVA, Sidak's. Data is presented as mean ± S.E.M or in box plots representing minimum, maximum, median, 25/75 percentile and mean (indicated by a marker). Source data are provided as a Source Data file.

Cytoplasmic N-terminal S15 phosphorylation of Cav2.3 has been previously detected in global phosphoproteomics screens from brain (phosphosite.org), but the responsible kinase and the role of this phosphorylation were unknown. N-terminal deletions in Cav2.3[60] or the related Cav2.2[86,87], result in channels with normal gating and conduction properties. Inactivation kinetics were not measured in these studies, yet interestingly Cav2.2 N-terminal deletion mutants seemed to have slower decaying currents, indicating that this region participates in the regulation of inactivation[86]. Analogously, loss of N-terminal Ser14/15 phosphorylation slows Cav2.3 inactivation in HEK293 cells and neurons, underscoring a previously overlooked regulatory role of the Cav2.3 N-terminus. This domain may interact with Cavβ, as previously suggested for Cav2.2[87].

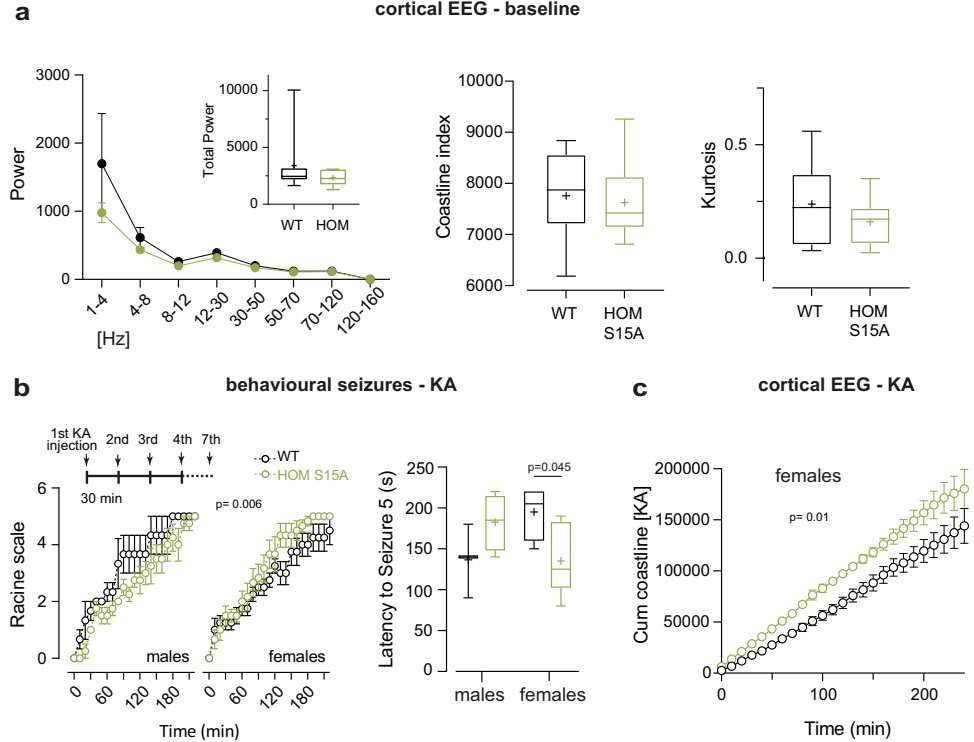

**Fig. 5 | Electrocorticogram features and seizure propensity in Cav2.3 phosphomutant mice. a** left: two-week baseline ECoG spectral analysis and total power (inset) in WT ($n = 8$) and HOMS15A ($n = 10$) 9–12 week-old mice; corresponding coastline index (middle) and kurtosis (right) ($p > 0.05$, Two-Way ANOVA and two-tailed unpaired $t$ tests). No sex differences were observed. **b** left: behavioural scoring of kainic acid (KA)-induced seizures (WT males $n = 3$, females $n = 4$; HOMS15A males $n = 4$, females $n = 6$; sex x genotype $p = 0.006$, Three-Way ANOVA) using repetitive 5 mg/kg i.p. injections (inset); right: latency to stage 5 tonic clonic seizures per gender (two-tailed unpaired $t$ test). **c** Corresponding cumulative ECoG coastline during KA seizures in females (WT $n = 3$, HOM S15A $n = 4$, time x genotype $p = 0.018$, Two-Way ANOVA). Data is presented as mean ± S.E.M or in box plots representing minimum, maximum, median, 25/75 percentile and mean (indicated by a marker). Source data are provided as a Source Data file.

We also find that cholinergic stimulation amplifies Cav2.3 GoF in absence of CDKL5 phosphorylation, by prominently shifting activation $V_{1/2}$ and increasing currents at maximal activation voltages, thus revealing a second molecular mechanism by which loss of pS14/15 enhances Cav2.3 function. It is known that muscarinic enhancement of Cav2.3 depends on Gαq signalling and PKC phosphorylation at multiple sites, including the intracellular loop I-II[62], the most prominent site of α1-β interaction[54]. We show that loss of pS14 interferes with PKC enhancement upon carbachol application, but the mechanisms of this interplay are yet to be established. We ensured multi-plasmid expression by co-transfecting GFP at lower concentration and our immunostainings verified widespread protein co-expression by in HEK293 cells (Supplementary Fig. 3). However, some of the variability in our data could be explained by cell-to-cell differences in levels of co-expressed plasmids.

Our recordings from hippocampal neurons in Cav2.3 S15A mice demonstrate that S15 regulates Cav2.3 function in neurons. We show that pS15 is critical for R-type current inactivation and regulation of firing by cholinergic stimulation. Isolating Cav2.3 currents is challenging due to space clamp limitations and the need for inhibitors of other voltage-gated channels. These drugs may have promiscuous effects on Cav2.3[88] or incompletely block other $Ca^{2+}$ currents[56]. We observed enhanced neuronal excitability and increased proepileptiform cholinergic ADPs/DPPs[59,64] in Cav2.3 phosphomutant mice, an effect that was not linked to carbachol suppression of AHPs or other intrinsic excitability properties in these cells. Instead, we suggest that the GoF of unphosphorylated Cav2.3 underlies the hyperexcitability seen in response to muscarinic stimulation. Interestingly, *Cdkl5* KO mice have altered cholinergic tone[76,89] and the muscarinic antagonist Solifenacin was shown to ameliorate network defects in

CDD patient-derived neurons[90], supporting an enhanced cholinergic modulation in this human model.

We also found behavioural similarities between Cav2.3 phosphomutant mice and *Cdkl5* KOs. Specifically, S15A mice of both sexes show reduced encoding and retention of fear memory, a robust phenotype replicated in full body KO[15,74,77,91] and excitatory neuron-specific[92] *Cdkl5* mouse models across multiple laboratories. On the other hand, Cav2.3 KO mice display enhanced contextual fear conditioning[38], the opposite of our gain-of-function mutants. Cav2.3 S15A also recapitulate to some extent the spatial cognitive deficits observed in *Cdkl5* models in the Barnes maze[74,77,78], the reduced home-cage locomotion[49,74] and altered sociability observed in global[15,49,73,74,79], kinase-dead knock-in[91] and inhibitory neuron-specific deletion[93] models. The deficits observed in voluntary wheel use in Cav2.3 phosphomutants could be indicative of mild hypotonia[94,95], and warrant further investigation.

Importantly, with the exception of fear conditioning, the absence of behavioural overlap between S15A and *Cdkl5* KO mice in each sex (locomotion, social, Barnes maze) as well as the lack of phenotype in phosphomutants for other robust cognitive and motors behaviours such as Y-maze, rotarod, limb clasping and hyperactivity, may be due to additional CDKL5 substrates involved in these behaviours in KO animals.

Cav2.3 KO mice exhibit reduced susceptibility to chemical absence seizures[96–98] and hippocampal KA-induced seizures[52]. In agreement with Cav2.3's role in epilepsy, we find increased seizure susceptibility in females in the low-dose KA injection paradigm in our Cav2.3 S15A mice. Overall, S15A behaviour and seizure phenotypes mirror to some extent human clinical manifestations of CDD, arguing that slowed inactivation and enhanced cholinergic modulation of unphosphorylated Cav2.3 may contribute to these symptoms.

The difference between males and females, observed in both ECoG and behavioural seizure analysis could reflect biological sex differences related to Cav2.3 function but requires further experiments with larger numbers of mice to ascertain. Nevertheless, sex-specific differences should be taken into consideration in future *Cdkl5* KO mouse behaviour and seizure susceptibility experiments. In *Cdkl5* KO males, KA-induced seizure susceptibility changes have not been observed[18,49], but this has not been explored in females. On the other hand, reduced latency to seizure initiation with pentylenetetrazole (PTZ, a GABA$_A$ receptor inhibitor) injections has been reported in *Cdkl5* KO males[91] as well as heterozygous females[75]. Finally, increased severity of seizures upon NMDA injections was observed in male *Cdkl5* KO mice due to increase NMDAR function[18], however NMDAR function is not altered in *Cdkl5* KO rats[28]. Disturbance-associated seizures are observed in aged heterozygous females in two different CDD mouse models[81,83], an occurrence that may be partly linked to changes in Cav2.3 regulation. It should be noted, however, that S15A gain-of-function in mice is not sufficient to observe spontaneous seizures.

In rodents, it is possible to reverse adult phenotypes with reintroduction of CDKL5 genetically[73] or using AAV viruses[79], indicating an open therapeutic window for treatment. Here, we report the functional regulation of Cav2.3 by CDKL5 phosphorylation, which impinges of neuronal excitability. Thus Cav2.3 GoF in absence of CDKL5 may be an important molecular mechanism in CDD pathology. Specific Cav2.3 inhibitors would be required to test if inhibition of Cav2.3 could ameliorate CDKL5 deficiency phenotypes in mouse and human models. In addition to CDKL5 and CACNA1E encephalopathies, increased Ca$^{2+}$ influx through Cav2.3 may contribute to seizures in Fragile X syndrome, as reduced FMRP function leads to increased Cav2.3 expression[99]. Cav2.3 also increases susceptibility to drug-induced Parkinson's disease[100,101]. We propose that Cav2.3 inhibitors/ modulators could be beneficial for CDD patients and potentially for a broader population of neurological patients in the future.

## Methods

### Mouse handling and mutant generation

Animals were bred and handled in accordance with the Animals (Scientific Procedures) Act 1986 of the United Kingdom and used following the "3Rs" principles. Protocols and procedures were approved by The Francis Crick Institute Ethical Committee, UCL Animal Welfare and Ethical Review Body and the UK Home Office. Animals were housed in a controlled environment with 12-hour light/dark cycle (20–24 °C, 55 ± 10% humidity) and fed ad libitum. Except for some in vivo experiments, mice were housed in groups. Mouse strains were backcrossed into C57 Bl/6 (Jackson) and none of the experimental mice were immunocompromised. Male and female mice aged embryonic day (E)16.5 to postnatal 40 weeks were used as specified.

*Cdkl5* KO animals were a kind gift from Cornelius Gross[49]. Mutant Cav2.3 Ser15Ala animals were created in-house using the CRISPR/Cas9 system. The guide contained the sequence 5'AGGCUCAGGCGAUGGAGACU3'. This replaced the original DNA sequence of 5' AGGCGCAGGCGATGGAGACT 3' in CACNA1E. Genetically altered C57Bl6 embryos were obtained using electroporation as part of Francis Crick Institute's Genetic Modification Services. Genotype was determined by in-house PCR or Sanger sequencing (Source Bioscience).

### SILAC phosphoproteomics sample preparation

Mouse cortical neurons were cultured from individual male E16.5 embryos from a heterozygous *Cdkl5* female. *Cdkl5* +/Y (WT) or −/Y (KO) genotype of embryos was determined afterwards. Neurons were plated at a density of $8 \times 10^6$ cells per 10 cm dish coated with 60 µg/ml poly-D-lysine (Sigma) and 2.5 µg/ml laminin (Sigma). Based on methods to achieve accurate SILAC ratios in non-dialyzed serum[102,103], cells were grown in neurobasal media free of L-Arginine and L-Lysine (Invitrogen) supplemented with either L-Lysine-8 (U-13C6, U-15N2) and

L-Arginine-10 (U-13C6, U-15N4) or L-Lysine-4 (4, 4, 5, 5-D4) and L-Arginine-6 (U-13C6)(Cambridge Isotope Laboratories). L-Proline (200 mg/ml) was supplemented to prevent Arginine to Proline conversion. Each sample was checked for adequate SILAC label incorporation (>90%) and minimal Arginine to Proline conversion (<10%). At DIV4, 1 µM Ara-C was supplemented to inhibit glia growth. For every embryo, one dish of cortical neurons was labeled with K8R10 (heavy) and one dish was labeled with K4R6 (light) amino acids to provide technical replicates. At DIV12, SILAC labelled primary neurons from 3 *Cdkl5* WT and 3 *Cdkl5* KO animals were lysed in lysis buffer containing 20 mM Tris pH7.5, 100 mM NaCl, 10 mM MgCl$_2$, 0.25% IGEPAL NP40, 0.5 mM DTT, 1x Protease Inhibitor Cocktail, 1 µM Okadaic Acid.

Samples were prepared for mass spectrometry as previously described in[104]. In brief, protein lysates were mixed in a 1:1 ratio to obtain 3x heavy WT/light KO and 3x heavy KO/light WT samples containing 1 mg total protein. Mixed samples were reduced with 5 mM DTT, alkylated with 10 mM iodoacetamide and quenched with 7.5 M DTT prior to o/n digestion trypsin at 37 °C. Peptides were desalted using 50 mg Sep-Pak C18 Cartridges (Waters) and vacuum dried completely before further digestion at 37 °C using Lys-C for 2 hrs, followed by o/n trypsin digestion. Peptides were desalted and dried again. Mixing and digestion checks were performed after each digestion step. Samples were enriched for phosphopeptides according to an in-house protocol using 5 mg of Titansphere titanium dioxide beads (GL Sciences) incubated for 10 min with peptides resuspended in loading buffer (80% acetonitrile, 5% trifluoroacetic acid, 1 M glycolic acid). TiO2 beads were subsequently washed with loading buffer, wash buffer 1 (80% ACN 1% TFA) and wash buffer 2 (10% acetonitrile, 0.2% trifluoroacetic acid), and peptides were finally eluted using 1% ammonium hydroxide and 5% ammonium hydroxide subsequently. Combined phosphopeptide eluates were vacuum dried completely and desalted using C18 Stage Tips and vacuum dried completely. Samples were stored at −80 °C until required for analysis by mass spectrometry.

### Mass spectrometry of enriched phosphopeptides

Samples were analysed by online nanoflow LC-MS/MS using an LTQ Orbitrap Velos mass spectrometer (Thermo Scientific) coupled to an Ultimate 3000 RSLCnano (Thermo Scientific). Resolubilised sample (10 µL per injection in 1% aqueous trifluoroacetic acid, TFA) was loaded via autosampler into a 20 µL sample loop and pre-concentrated onto a cartridge trap column 300 µm I.D. × 5 mm, packed with Acclaim Pep-Map100 C18, 5 µm, 100 Å using the loading buffer, 2% v/v acetonitrile, 0.05% v/v trifluoroacetic acid, 97.95% water (Optima grade, Fisher Scientific) at a flow rate of 20 µL/min for 2 min in the column oven held at 40 °C. Peptides were backflushed and gradient eluted onto a Pep-Map RSLC C18 75 µm × 50 cm, 2 µm particle size, 100 Å pore size, reversed phase EASY-Spray analytical column (Thermo Scientific) at a flow rate of 250 nL/min and with the column temperature held at 40 °C, and a spray voltage of 2 kV using the EASY-Spray Source (Thermo Scientific). Gradient elution buffers were A 0.1% v/v Formic Acid, 5% v/v DMSO, 94.9% v/v water and B 0.1% v/v Formic Acid, 5% v/v DMSO, 20% v/v water, 74.9% v/v acetonitrile (all Optima grade, Fisher Scientific aside from DMSO, Honeywell Research Chemicals). The gradient elution profile was 2% B to 30% B over 143 min, then 30% B to 50% B over a further 20 min. The mass spectrometer was operated in data dependent acquisition mode with the top 10 most abundant peptides selected for MS/MS by either collision-induced dissociation, multi-stage activation or higher-energy collisional dissociation fragmentation techniques. The three instrument methods used a capillary temperature of 275 °C, an MS1 Orbitrap scan resolution of 60,000 FWHM at m/z 400, mass range 300–2000 m/z, S-Lens RF level 60%, FTMS Full AGC target 1e6, maximum injection time 500 ms and spectra were acquired in profile. Only precursors with charge state >2 were permitted for selection for fragmentation and dynamic exclusion

was enabled to exclude after $n = 1$ times within 20 s for 20 s. Fragmentation was subsequently performed on all selected precursor masses and the MS2 scan data was acquired in centroid mode. For the method with CID activation MS2 scans were acquired in the linear ion trap following CID fragmentation with normalized collision energy of 30%, Ion Trap MSn AGC target 1e4 and maximum injection time 100 ms. For the method with CID activation and multi-stage activation enabled parameters were as for CID but with normalized collision energy of 35% and a neutral loss mass list (32.70, 49, 98). For the method with HCD activation MS2 scans were acquired in the Orbitrap at a resolution of 7500 FWHM at m/z 400, following HCD fragmentation with a normalized collision energy of 45%. The parameters used for the HCD MS2 scan were fixed first mass 100 m/z, FTMS MSn AGC target 5e4, maximum injection time 500 ms.

The acquired raw mass spectrometric data was processed in MaxQuant[105] (version 1.3.0.5) for peptide and protein identification, the database search was performed using the Andromeda search engine against the *Mus musculus* canonical sequences downloaded from UniProtKB (August 2012, 77938 sequences). Default search settings were used including an FDR of 1% on the phosphosite, peptide and protein level and matching between runs for peptide identification. K4R6 and K8R10 were set as labels with a maximum of 3 labeled AAs. Phosphorylation at serine, threonine or tyrosine residues, oxidation (Met) and protein N-terminal acetylation were set as variable modifications. Carbamidomethylation of Cys-residues was set as a fixed modification.

Data was further processed in Perseus v.1.6.15.0. Phospho (STY) sites from the six experimental samples in the MaxQuant search were loaded and site tables expanded. Potential contaminants and reversed hits were removed. Normalized H/L ratios were log2 transformed and filtered for rows containing at least three valid values. Potential CDKL5 substrates were annotated using the R-P-X-pS/pT motif. A one-sample t-test was performed comparing the normalized ratios to the null hypothesis and visualized as a volcano plot.

The mass spectrometry proteomics data have been deposited to the ProteomeXchange Consortium via the PRIDE[106] partner repository with the dataset identifier PXD038505.

## Mutagenesis

Point mutations were introduced using QuickChange site-directed mutagenesis (Agilent). Complimentary primers contained the desired mutations flanked by at least 18 bp. To ensure mutagenesis efficiency for each construct, two separate PCR reactions with each primer were set up: 0.2 µM primer, 100 ng template DNA, 0.2 mM dNTP mix, 1x Pfu buffer and 2.5 U Pfu Ultra HF. The PCR programme was run 4x, after which complementary samples were combined and the PCR run repeated 20x. PCR products were treated with DpnI for 1 h at 37 °C and transformed into XL-10 Gold competent cells. At least 4 colonies were used for each DNA miniprep (QIAprep, Qiagen) and mutations were confirmed by Sanger sequencing (Source Bioscience). The constructs used were as follow: (1) HA-tagged human α1E subunit (GenBank L27745.2, gift from L. Parent, Université de Montréal), in a commercial vector under control of CMV promoter[107]. (2) The N-terminal FLAG-tagged full-length human CDKL5[107] and N-terminal HA-tagged human CDKL5$_{1-352}$ kinase domain construct kinase as previously described[20].

## Cell line maintenance and transfection

Tetracycline-inducible HEK293 cells expressing human Cavβ3 and α2δ1 subunits were obtained from SB Drug Discovery (Glasgow, UK). HEK293 cells stably expressing human Cav β1b and α2δ1 subunits were kindly provided by Andrew Powell (GSK Ltd., Middlesex, UK).

All cells were cultured in high glucose DMEM with 10% Fetal Bovine Serum (tetracycline free, Clontech or Gibco) and penicillin/streptomycin 50 units/ml-50 mg/ml. They were passaged using TrypLE Express and maintained under selection antibiotics as appropriate:

HEKβ3/ α2δ1, Zeocin 300 mg/ml (Invivogen) & Blasticidin 5 mg/ml (Invivogen); HEKβ1b/α2δ1, Puromycin 1 mg/ml and Hygromycin-B 200 mg/ml. Cell culture reagents were obtained from Gibco/Invitrogen unless specified.

Selection antibiotics were removed before transfection and doxycycline 1 µg/µl added in the case of β3-expressing cells. Transient transfection was carried out using X-tremeGENE™ 9 reagent (Roche) following manufacturer's instructions. Total DNA was 1.5–1.6 µg. For Western blot experiments, human α1E-HA subunit and CDKL5 (HA-kinase domain or FLAG-full length) were co-transfected in a 1:1 ratio, substituting kinase for empty pcDNA3 vector in some cases. For electrophysiological recordings α1E-HA, FLAG-CDKL5 full length and GFP (pcDNA3) were co-transfected at a ratio of 10:4:1 (HEK β3 cell line) or 8:4:1 (HEKβ1b cell line). To study muscarinic regulation, human α1E-HA, FLAG-CDKL5, muscarinic receptor type 3/1 (CHRM3/1, Addgene) and GFP were co-expressed at a ratio of 8:4:3:1. For PKC experiments, some recordings were performed in cells transfected with human α1E-HA, FLAG-CDKL5, M3/M1-IRES-mcherry at a ratio of 1:0.7:0.7. This vector was cloned in-house by inserting the IRES-mcherry sequence downstream of the muscarinic receptor using XhO1/Xba1 sites. For dopamine modulation experiments, human α1E-HA, FLAG-CDKL5 and dopamine D2 receptor (gfp-DRD2, Addgene) were co-transfected at a ratio of 2:0.75:1.

## Immunofluorescence and confocal microscopy

HEK cells were split 24 h post transfection, re-plated onto glass coverslips and allowed to recover overnight. Cells were then fixed for 10 min in PFA/sucrose, washed in PBS and stored at 4 °C until use. For immunocytochemistry experiments, cells were treated with blocking and permeabilization buffer (10% goat serum and 0.01% Triton X in PBS) for 1 h before primary antibody incubation overnight at 4 °C. Primary antibodies were as follow: rat or mouse anti HA (for Cav2.3 detection 1:500–1000; Roche 11867423001 clone 3F10 or Biolegend 901513 clone 16B12), mouse anti CDKL5 or rat anti-FLAG (for CDKL5 detection, 1:1000, Santa Cruz sc-376314 clone D12 or Thermo Fisher MA1-142 clone L5), rabbit anti M3 (1:1000, Alomone AMR-006), anti GFP sdAb FluoTag-X4 (NanoTag Biochecnologies N0304or chicken anti GFP (1:1000, Aves GFP-1020). Secondary antibody incubation was for 1 h at RT. These were as follow: donkey anti rat Cy3 (Jackson 712-165-153), donkey anti-mouse 647 (Jackson 115-605-003), goat anti-rabbit 405 (Thermo Fisher A48254), goat anti-chicken 488 (Thermo Fisher A32931). In some experiments nuclei were stained with DAPI 1:5000 (Thermo Fisher A48254). Coverslips were mounted on slides with Fluoromount-G (Southern Biotech 0100-01) and imaged the following day at the earliest. All images were acquired with a Zeiss Invert 880 confocal microscope either as single images or Z-stacks (8–10 µm at 1 µm interval) using the 40x oil immersion objective. Z-stack images were maximally projected to generate example pictures. Matching transfection conditions were imaged on the same day using identical confocal imaging parameters.

## Western blotting

Three different types of samples were used for Western blot experiments: (1) HEK293 cells 48 h post-transfection, (2) Mouse brain tissue collected after cervical dislocation, followed by dissection and snap-freezing in liquid nitrogen, (3) Human iPSC derived neurons obtained as frozen pellets from Cleber Trujillo (UCSD, USA). Protocols for iPSC generation, patient *CDKL5* mutations and related control details are described in[90]. All lysates were prepared in 1x/2x sample buffer (Invitrogen) with 0.1–0.2 M dithiothreitol (DTT, Sigma) or RIPA buffer with phosphatase and protease inhibitors (later diluted in sample buffer plus DTT). This was followed by sonication, centrifugation at 13,000 g and denaturation at 70 °C, 10 min. Protein was loaded onto NuPAGE 8% Bis-Tris polyacrylamide gels (Invitrogen) for electrophoresis and subsequently transferred to polyvinylidene difluoride membranes

(Millipore) for 15–22 h at 20 V in 10% MetOH, Tris Glycine buffer. Membranes were blocked for 30 min in 5–10% skimmed milk and incubated with primary antibodies overnight at 4 °C or 1 h at room temperature (RT). Secondary incubation in horseradish peroxidase-conjugated (HRP) antibodies was for 1 h at RT. Chemiluminescence signals were detected using Amersham ECL (Cytiva) and Amersham Imager (GE Healthcare). Analysis was performed with Fiji 2.1. Phospho-S14/15 channel levels were measured as the ratio between pS14/15 Cav2.3 signal and total Cav2.3 signal. Total channel levels are expressed as the ratio over loading controls GAPDH or Tubulin. Each replicate in a blot constitutes a data point normalized to the average internal control signal in each blot. At least two independent immunoblot experiments with technical replicates were used for quantification, as specified in figure legends. All uncropped and unprocessed scans used in this manuscript are provided in the accompanying Source Data File.

Primary antibodies used in each sample were: mouse anti-HA 1:2000 (Biolegend 901513 clone 16B12, HEK293), rabbit anti pS15 Cav2.3 1:500 (Covalab custom, HEK293, mouse brain & human neurons), mouse anti Cav α1E 1:1000 or 1:2000 (Synaptic Systems 152411 clone 62C10, mouse brain and human neurons), rabbit anti CDKL5 1:2000 (Atlas HPA002847, HEK293), mouse anti GAPDH 1:50000 (Abcam ab8245 clone 6C5, mouse brain & human neurons), mouse anti αTubulin 1:100000 (Sigma T9026 clone DM1A, mouse brain). The custom made Cav2.3 Ser15 phosphospecific antibody was raised by immunizing rabbits with peptide PRPG(pS)GDGDSDQSRNC. Phosphorylated Cav2.3 antibody was obtained after double purification. First, a control antibody is selected against non-phosphorylated peptide PRPGSGDGDSDQSRNC. Next the eluted fraction was purified again by binding to PRPG(pS)GDGDSDQSRNC -linked beads, obtaining pSer15 Cav2.3 (custom). Secondary antibodies used were: donkey anti rabbit HRP 1:10000 (Jackson 711-035-152), donkey anti mouse HRP 1:10 000 (Jackson 715-035-151).

## In vitro electrophysiology

**HEK293 cells.** Cells were split 24 h after transfection, replated on glass coverslips and allowed to recover overnight. Whole-cell patch clamp recordings in isolated cells were performed 48–72 h post-transfection at room temperature. Transfected cells were identified by GFP fluorescence. Data were sampled at 20 kHz and filtered at 1–2 kHz. Cells were continuously perfused (1 ml/min) with extracellular solution containing (in mM): NaCl 120, TEACl 20, HEPES 10, glucose 10, KCl 5, MgCl$_2$ 1, CaCl$_2$ or BaCl$_2$ 5, pH 7.4 with NaOH. Pipettes (2.5–3.5 ΩM) were pulled from borosilicate glass (1.2 × 0.69 mm, Harvard Bioscience) and filled with the following intracellular solutions adapted from[41] to minimize current run-down (in mM): CsMeS 140, EGTA 5, MgCl$_2$ 0.5, MgATP 5, HEPES 10, for GPCR experiments; CsMethanesulfonate 125, TEACl 5, EGTA 5, MgATP 5, Na$_3$GTP 0.3, Na-phosphocreatine 5, Na-pyruvate 2.5, HEPES 10, pH 7.25, 285–295 mOsm, all other recordings. Upon seal break, currents were evoked every 5/10 s with a step from −80/−100 mV (holding potential, HP) to 0/+10 mV. After an initial run-up period, steady state current voltage (IV) relationships were obtained with 100 ms depolarising test steps to +60/+80 in 10 mV increments every 10 s. For voltage dependence of inactivation the protocol consisted of a 1 s pulse from −140 to +20 mV in 10 mV increments, followed by a 50 ms test pulse to 0 mV, every 10 s. Peak currents ($I_{peak}$) in response to test voltages ($V_{test}$) were used to plot IVs and measure the voltage dependence of activation and inactivation. Reversal voltage ($E_{rev}$) was estimated by extrapolating a linear fit to the IV curve from +20-+40 mV. Conductance (g) was calculated using the equation ($g = (I_{peak})/(V_{test}-E_{rev})$). Voltage of half-maximal activation or inactivation ($V_{1/2}$) were measured by fitting with a Boltzmann equation ($Y = A + (B-A)/(1+exp((V-V_{1/2})/K))$) where Y is the conductance or the current, and $A$ and $B$ are the minimum and maximum amplitudes of the fit. Time course of open state inactivation ($\tau_{inact}$) was estimated by fitting a single exponential function to current decay in response to

test pulses. The average of 3 fits at each voltage was used. For all transfection conditions, cells with outward currents >40% of the peak current at +10 mV or with $\tau_{inact}$ that did not decay uniformly with depolarization were excluded from analysis. Similarly, cells were not used for current density or $\tau_{inact}$ comparisons if run-down was >30% of the maximal current recorded in the experiment as this is known to affect inactivation time[41]. To examine recovery from inactivation, an inactivating pulse to +10 mV of 0.8–1 s (pulse 1), was followed by a variable recovery period (Δ150–200 ms) and a test pulse to +10 mV (pulse 2), at 0.1–0.3 Hz. Time course of recovery was calculated using the ratio of $I_{peak}$ in response to pulse 2/pulse 1 plotted against duration of recovery period and fitted with single exponential equation. Carbachol, quinpirole and PMA were bath applied at ~3 ml/min whilst activating Cav2.3 currents with brief monitoring (25 ms) test steps to 0 or +10 mV, every 2, 5 or 10 s, depending on speed of run-down. Upon steady state, IVs were obtained. Cells included in the analysis showed no significant changes in series resistance and a consistent shift in activation $V_{1/2}$, with current amplitude matching between IV and brief monitoring steps. Leak and capacitance subtraction was applied to all recordings using -P/5 protocol and series resistance compensated from 75–95% to keep voltage error <7 mV.

## Ex vivo electrophysiology

**Hippocampal slices.** Mice P10-11 or P35-42 were anaesthetized by intraperitoneal injection of ketamine (80 mg/kg) and xylazine (10 mg/kg). To avoid bias, the genotype was unknown at the time of experiment. Animals were decapitated and the brain transferred to ice cold artificial cerebrospinal fluid (standard aCSF) containing (in mM): NaCl 125, NaH$_2$PO$_4$ 1.25, NaHCO$_3$ 26, KCl 2.5, glucose 25, MgCl$_2$ 1, CaCl$_2$ 1, saturated with carbogen. Cerebellum and olfactory bulb were discarded. Coronal or transverse hippocampal slices (300 μm) were prepared with a Leica VTS 1200 S vibratome and transferred to an immersion storage chamber containing bubbled aCSF at 35 °C for 30 min. Slices were allowed to recover for a further 30 min at room temperature (RT) before electrophysiology experiments. Somatic whole-cell patch clamp recordings were obtained from CA1 pyramidal neurons using borosilicate electrodes (3–4.5 MΩ). Series resistance ($R_s$) and input resistance ($R_{in}$) were monitored throughout recordings using a 100 ms, −5 mV step from −60 mV to evoke passive membrane responses. $R_s$ was calculated from the amplitude of the capacitive transient and $R_{in}$ from the steady state currents. These were sampled at 20 kHz and filtered at 10 kHz. The maximum change in Rs permitted was 25%.

**R-type Ca$^{2+}$ currents.** Recordings were obtained from young neurons (P10-11) no longer than 6 h post slicing using the blind technique. Currents were recorded at RT from P10-11 coronal slices during bath application of a modified aCSF solution (in mM): NaCl 115, TEACl 10, NaHCO$_3$ 26, KCl 2.5, NaH$_2$PO$_4$ 1.25, glucose 25, MgCl$_2$ 1, CaCl$_2$ 2; and in presence of the following inhibitors (in μM): TTX 0.5, gabazine 1, APV 25, NBQX 5, 4AP 5, Nifedipine 10 (Tocris), w-conotoxin GVIA 2, w-agatoxin IVA 0.2, w-conotoxin MVIIC 2 (Alomone) and BSA 1 mg/ml. Drug stock solutions were prepared in water and stored at −20 °C. External solution was constantly bubbled and re-circulated. Slices were exposed to ion channel inhibitors for at least 20 min before voltage clamp acquisition. Patch pipettes were filled with (in mM): CsMethanesulfonate 130, TEACl 10, Na-phosphocreatine 5, MgATP$_2$ 4, Na$_3$GTP 0.3, HEPES 10, EGTA 2 and biocytin 0.2% pH 7.3 with CsOH. After 10 min stabilization in the whole-cell configuration, IV curves were obtained with voltage steps from −80 to +40, HP = −70 mV. A -P/5 leak and capacitance subtraction protocol was used and $R_s$ was compensated 65–85%, to minimize voltage error. Capacitance ($C_m$) was calculated using equation: $C_m$ [pF] = $\tau_m$ [ms]/$R_s$[MΩ], where $\tau_m$ is the decay time constant of capacitative transients. Time course of open state inactivation was estimated by iterative single exponential fit to

current decay. A positive correlation was found between $\tau_{inact}$ and $C_m$ for $C_m < 80$ pF, thus this was established as the cut off cell size criteria for inclusion in $\tau_{inact}$ comparisons. Cells with large outward currents at depolarising voltages (>50% of the peak current at +10 mV) were also excluded.

**Small-conductance Ca$^{2+}$-activated (SK) currents and excitatory postsynaptic currents (EPSC).** Recordings were obtained from adult neurons (5–6 weeks) in transverse slices. SK currents were measured as tail currents following a 100 ms step from −50 mV to +10 mV (0.3 Hz) to engage VGCCs. Standard aCSF (CaCl$_2$ 2 mM, RT) was supplemented with (in μM): TTX 0.5, TEA 1000 and XE991 5, to isolate SK currents by inhibiting temporally overlapping voltage-gated Na$^+$ and K$^+$ conductances. The intracellular solution was (in mM): KGluconate 135, KCl 10, Na-phosphocreatine 10, MgATP$_2$ 2, Na$_3$GTP 0.3, HEPES 10, pH 7.3, supplemented with 8-cloro-phenylthio cAMP 50 μM, to inhibit the slow afterhyperpolarizing current. SK current identity was confirmed at the end of all experiments by reversible d-tubocurarine (dTC, 50 μM) inhibition. dTC-subtracted currents were used for quantification. Traces were sampled at 5 kHz and filtered at 1 kHz. SK current amplitude was measured at the peak of the after-current and charge as the integral 500 ms post stimulus. EPSCs were isolated at −70 mV in standard aCSF (1) (CaCl$_2$ 2.5 mM, 30–32 °C) with gabazine 1 μM and Kgluconate-based internal solution. Traces were sampled at 10 kHz and filtered at 2 kHz. Upon 5 min stabilization in whole-cell, a 2 min gap free trace was analysed using template-based event detection and threshold match >3.

**Current clamp recordings.** Recordings were obtained from adult neurons (5–6 weeks) in coronal slices in standard aCSF (2 mM Ca$^{2+}$, 30–32 °C) and Kgluconate-based internal solution. Membrane potential ($V_{rest}$) was measured throughout the recording with 0 current injection and maintained around −65 mV during stimulation. Input-output relationships were obtained with 1 s current injections from −100 to 450 pA in 50 pA increments. Traces were sampled at 10 kHz and filtered at 5 kHz. Action potentials were identified with threshold-based event detection (0 mV) and amplitude measured from $V_{rest}$. Spike attenuation during the train was calculated as ratio between last and first spike amplitudes. Depolarizing plateau potentials (DPPs) were defined as sustained depolarizations >20 mV upon stimulus termination. Medium duration afterdepolarizations (ADPs) were measured 100 ms post-1s step depolarization. Firing patterns were compared using binomial tests or comparisons of non-linear least squares regression fits to the data distribution. The effect of carbachol on the medium and slow afterhyperpolarizations (mAHP and sAHP) was used as positive control for muscarinic receptor activation. AHPs were evoked at −65 mV by a burst of five 2 nA, 2 ms somatic current injections at 100 Hz every 20 s and measured at the peak and 500 ms post stimulus respectively.

All patch clamp data was obtained with Multiclamp 700B, Digidata 1440 A and pClamp$^{TM}$ 10 software (Molecular Devices). Data were analysed using Clampfit 10.7/11.2 and OriginPro 9.8. Liquid junction potential was not corrected.

**In vivo electrophysiology**
**Chronic electrocorticogram recordings.** Surgical procedures were performed in adult male and female mice (8–10 weeks old) using a stereotaxic frame (Kopf) under isoflurane anaesthesia and temperature control. Each animal was subcutaneously implanted with an electrocorticogram (ECoG) transmitter (A3022B-CC-B45-B, Open Source Instruments, Inc.). The recording electrode was placed above the visual cortex (AP -2, ML 1.5) and the ground electrode in the contralateral prefrontal hemisphere (AP 1.8, ML 1.5). Implanted WT and Hom S15A animals (total 18) were housed individually and no drug treatment was given. The ECoG was recorded wirelessly (sampling

frequency 512 Hz, band-pass filter 1–160 Hz) for two weeks using software from Open Source Instruments, Inc. Simultaneous video recordings were obtained (6x/hour) using IP cameras from Microseven (https://www.microseven.com/index.html). The coastline, kurtosis and power spectrum analysis (1–160 Hz) were performed using Python. Researchers were blinded to animal genotype during data acquisition and analysis.

**Kainic acid susceptibility.** At the end of the chronic recordings, low dose 5 mg/kg kainic acid (Tocris Bioscience) dissolved in sterile saline was administered intraperitoneally every 30 min to assess brain susceptibility to increasing dosage of chemo convulsant. The clock started at the first injection from 0 min. Seizure severity was initially assessed while the experiment was ongoing at 10-minute intervals, using a modified Racine scale: 1. immobility; 2, hunched position with facial jerks, 3. rearing and forelimb clonus, 4. persistent rearing and forelimb clonus, falling, 5. generalized tonic clonic convulsions, or wild jumping[85]. These assessments were confirmed by re-analyzing the video recordings. Latency measures the length of time from the start of the experiment to the first indication of generalized seizure (Stage 5 on Racine scale). Both kainic acid inductions and analysis were performed by a researcher blinded to the genotypes. A total of 17 implanted mice were used for susceptibility experiments, one male mouse was removed from the study due to surgical complications with the transmitter. In a subset of these where transmitter performance was not compromised, ECoG recordings during seizure inductions were recovered and used for quantification. Traces were analysed using semi-automated seizure detection as described previously[84]. The cumulative coastline was calculated from when the first epileptiform activity was detected until the mice reached generalized seizure (stage 5).

**Behavioural analysis**
**Home Cage monitoring.** Two separate cohorts of WT and homozygous S15A Cav2.3 mice of both sexes and aged between 12–16 weeks were used. Animals were housed individually and monitored using the Digital Ventilated Cage system (DVC®,Techniplast, Italy). The DVC consists of standard size, barcoded IVC units equipped with a 3 × 4 arrangement of cage-floor electrodes. Capacitance sensing-based detection of animal position allows for real-time tracking of locomotion activity 24/7. Following a 2-week-acclimatization period, baseline activity was recorded and analysed. Rotating wheels (Ø 4.4 in) were then introduced, and animals allowed to familiarize themselves for another week before data collection. All activity metrics were calculated using the DVC Analytics 3.4.0 platform[108]. For locomotion index, the average signal of all 12 cage electrodes was used. Data were grouped in 60 min bins for each day. Animals were kept in DVC cages up to the age of 40 weeks.

**Behaviour tests.** Behaviour experiments were conducted in 8-week-old WT and homozygous S15A Cav2.3 of both sexes, housed under an inverted light cycle. Mice were handled daily for one week prior to experiments. Two cohorts of mice were used for all 7 behaviour tests, carried out in the specific order described below to minimise the possibility of one test influencing the evaluation of the subsequent test. Similarly, to avoid unnecessary stress, mice were allowed to habituate to the testing room for at least an hour before the behaviour tests. Lighting was manipulated depending on the specific behaviour experiment. Aversive light levels were avoided where possible. Animal weight was monitored weekly to check that it did not drop below 10% of the weight at the start of testing. All tests were video recorded for off-line analysis. Tracking data were analysed using Ethovision XT 15 (Noldus Information Technology, Netherlands) equipped with three-point (nose, body centre, and tail) detection settings.

**Open field test**. The apparatus consisted of an illuminated (70 lux) white floor PVC foam arena (50 × 50 × 40 cm). A central zone (16 × 16 cm) was pre-defined. Test mice were placed in the centre of the open field and allowed to explore the arena for 30 min. At the end of each trial, the mouse was returned to the home cage and the arena was cleaned. Total distance travelled in 30 min and percentage of time spent in the central zone were used as measures of locomotor activity and anxiety levels, respectively.

**Accelerating rotarod test**. Day 1: Mice walked on a rotating rod (Ugo Basile model 47650, Italy) at constant speed (4 rpm) for three minutes for two acclimatization trials. Day 2: Each mouse was placed on the rotating rod for three test trials, during which the rotation speed gradually increased from 4 to 40 rpm within four minutes. The inter trial interval was 1 h. Performance was evaluated by measuring the latency to fall.

**Hind-Limb clasping**. Mice were suspended by the middle of the tail and lifted 15 cm above the ground; the extent of hind-limb clasping was recorded for 15 s. The room lighting was kept at 90 Lux. Hind-limb clasping was scored from 0 to 3: score 0 = both hind limbs were splayed outward away from the abdomen; score 1 = one hind limb retracted inward toward the abdomen for at least 50% of the observation period; score 2 = both hind limbs partially retracted inward toward the abdomen for at least 50% of the observation period; score 3 = both hind limbs completely retracted inward toward the abdomen for at least 50% of the observation period. Hind-limb extension reflex severity scores were calculated by averaging three trials separated by 1 min.

**Three-chamber social interaction test**. The apparatus consisted of a white rectangular PVC foam arena (62 × 42 × 23 cm) divided into one central and two adjacent compartments of equal size compartments connected by square openings. Empty cylindrical wire cages (Ø 8 cm, height 18 cm) were placed in the lateral chambers. Each mouse was placed in the central chamber and allowed to explore the arena for 10 min. The brightness of the room was kept at 10 lux. Chamber occupancy during habituation was equal between groups. After this, the mouse was returned to the waiting cage for 3 min. To assess social preference, an unfamiliar WT mouse of the same age, sex and strain (stranger 1) was gently introduced inside one of the wire cages to serve as a social stimulus. An unfamiliar and inanimate object was added to the other wire cage as the non-social stimulus. The test mouse was placed in the apparatus containing the social and non-social stimuli for 10 min. After that, the test mouse was removed and kept in the waiting cage for 3 min. All compartments were cleaned with 30% ethanol solution before the next test phase. To assess preference for social novelty, the non-social object from the previous test phase was replaced by an unfamiliar mouse (stranger 2) and the previous stranger 1 (now familiar mouse) was placed into the wire cage of the opposite compartment. The test mouse was placed in the central chamber and allowed to explore the arena for the next 10 min, after which it was returned to the home cage. The location of the novel and familiar mice with respect to the side compartments was counterbalanced across trials. The exploration of the target mouse/object was scored when the mouse's nose was detected within 2 cm from the cylindrical wire cage.

**Y-maze spontaneous alternation test**. Testing occurs in a Y-shaped maze with three arms (1, 2, 3) made of opaque plastic (34 × 9 × 14 cm) oriented in a 120° angle. Each mouse was randomly placed into one arm and allowed to explore the maze for 10 min. Room lighting was kept at 10 lux. An arm entry was recorded manually when the mouse moved beyond the central triangle of the maze and entered an arm with all four paws. Alternation behaviour was defined as consecutive entries into each of the three arms in overlapping triplet sets (e.g.: 1, 2,

3 or 2, 1, 3 or 3, 2, 1). The percentage of alternations was calculated as the number of actual alternations divided by the maximum possible number of arm entries.

**Barnes-maze test**. Twenty-four hours before training, the animal was habituated to the apparatus and escape box for a minute. During the training phase (4 days), mice were trained to locate the target hole (with an escape chamber underneath) from among 20 holes evenly spaced around the perimeter of an elevated circular open field (Ø 96 cm). Each animal was initially placed in the centre of the arena covered by an opaque cylinder, which was removed 10–20 s after the start of the trial. Room lighting was kept at 635 lux. The mouse is then free to explore the platform for 3 min using four visual cues to aid navigation. If the mouse does not enter the escape chamber within 3 min the experimenter guides the mouse gently to the escape box (20 × 11 × 7 cm) and leaves the mouse inside for 1 min. This step is repeated two more times a day (3 trials in total per day) with at least 20 min in between where the mouse is placed back in its home cage. To assess learning, a probe test in carried out 24 h after the last training, where the escape box has been removed from the target hole. The mouse is placed on the platform and free to explore it for 3 min. Distance to first: distance walked from the centre of the arena to the target hole (centre-point detection); latency to first: time to reach the target hole from the centre of the arena (centre-point detection); errors to first are defined as checking any hole before reaching the escape box (nose-point detection).

**Contextual fear conditioning**. The fear conditioning chamber (27 × 27 × 35 cm, Ugo Basile, Italy) was equipped with a shock grid floor and a digital Near Infrared Red Video Fear Conditioning system. Mice from each genotype and sex were examined in four successive phases comprising: conditioned acquisition (day 1), memory of the conditioned background context A (day 2), exploration in new background context B (day 3) and memory extinction in the conditioned context A (day 4–7).

Context A was characterized by a cubic shape, illuminated at 100 Lux and the presence of a vanilla odour. The floor consisted of metal rods, mediating the foot shock. Context B was also cubic shaped, not illuminated and without additional odour cues. On day 1: Mice were introduced in the conditioning chambers scented with vanilla odour for 8 min and received 3 unconditioned stimuli (US: 1 s, 0.25 mA foot shock, 2 min inter-trials interval). To examine the conditioned response to the context, mice were reintroduced in the training context 24 h later (day 2) and monitored for 5 min in the absence of the US. On day 3, the mice were introduced to a new context (context B) for 5 min to examine whether freezing behaviour is associated to a specific context (context A). During days 4–7 mice were placed back in context A without US to assess extinction of the association of the specific context A and the US (which is described as a form of new learning).

## Statistical analysis

Statistical analyses were performed in GraphPad Prism 9. Two-tailed Student's *T* tests and One- or Two-Way ANOVA with Geisser-Greenhouse correction and post-tests were used for most statistical comparisons, unless otherwise specified. Repeated measures ANOVA was used as appropriate if there were no missing values. Data are presented as mean ± S.E.M. Box and whiskers (min to max) or violin plots are used to illustrate data spread and frequency distribution respectively; percentile 25/75, median and mean are indicated by lines/ markers. For behaviour tests, a post-hoc power analysis was conducted using the G*Power analysis tool[109]. Two- or Three-Way ANOVA were used to test sex or genotype effects as specified in figure legends.

## Reporting summary

Further information on research design is available in the Nature Portfolio Reporting Summary linked to this article.

## Data availability

The quantitative proteomics dataset generated from WT and *Cdkl5* KO mouse cultures has been deposited to the ProteomeXchange Consortium via the PRIDE partner repository under identifier PXD038505. The rest of the datasets generated and analysed during this study are provided in the Supplementary information and a Source Data File. Source data are provided with this paper.

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

## Acknowledgements

We thank Ultanir lab members for valuable discussion and thoughtful comments. We thank Helen Flynn for valuable assistance with mass spectrometry. We thank Richard Kraus (Merck & Co.) for helpful comments on the manuscript. We thank Crick GEMS team for creation of CACNA1E S15A mice. This work was supported by the Francis Crick Institute, which receives its core funding from Cancer Research UK (CC2037), the UK Medical Research Council (CC2037) and the Wellcome Trust (CC2037); Loulou Foundation Project Grant (11015) to S.K.U.; Crick-MSD Framework collaboration grant (11202) S.K.U. and M.S.C. For the purpose of Open Access, the authors have applied a CC BY public copyright licence to any Author Accepted Manuscript version arising from this submission.

## Author contributions

M.S.C., L.L.B., G.L. and S.K.U. designed the experiments. M.S.C. performed and analysed electrophysiology, biochemistry and immunocytochemistry experiments. L.L.B. performed mass spectrometry. Y.Q. and G.L. conducted/analysed EEG and kainic acid-induced seizure experiments. A.T.L. and M.S.C. performed behaviour experiments/analysis. L.S., S.M., S.C. provided technical assistance in molecular biology and electrophysiology. M.S.C. and S.K.U. wrote the initial version of the manuscript. J.R. contributed to the original draft and review/editing along with all authors.

## Funding

## Competing interests

The authors declare no competing interests.
