## [Peer Review File · Nature Communications]

Epilepsy-linked kinase CDKL5 phosphorylates voltage-gated calcium channel Cav2.3, altering inactivation kinetics and neuronal excitabilityREVIEWER COMMENTS

Reviewer #1 (Remarks to the Author):

This review was done by Holger Lerche from Tübingen University, Germany. The authors present comprehensive evidence in experiments ranging from molecular to both mouse and human cellular and whole animal levels, that Cav2.3 is a target of CDKL5 and that CDKL5-related DEE is at least partially caused by the lack of CDKL5-related phosphorylation of Cav2.3 and also a functionally related enhanced cholinergic stimulation of non-phosphorylated Cav2.3 channels. It is an impressive amount of work adding important novel information to the field. The manuscript is very clearly written with nice and clear figures illustrating the results. The methodology is sound except some minor issues. I have only few comments which are outlined as follows.

Comments

Line 52: ...aetiology OF approximately...

Line 88: Ref 44 does not provide functional studies on Cav2.3 channels, so should be cited only for confirming genetic findings in CACNA1E (as in line 87 and later).

The overlap between phenotypes of patients with CACNA1E- and CDKL5-related DEE, and also the differences, should be given in more detail. Maybe a supplemental table could be provided to give an easily accessible overview of both similarities and differences of both diseases. This should be also part of the discussion

It should be explicitly mentioned in the text, that S14 and S15 are corresponding serine residues in human and murine Cav2.3 channels.

Lines 167 and following paragraph: The authors should discuss and include additional evidence, that all three proteins (Cav2.3, CDKL5, mAChR) are expressed in their transfected HEK cells. I appreciate that there was an additional measurable functional effect of mAChR-related phosphorylation (hyperpolarizing shift of the activation curve), but it is not clear if all three proteins were really expressed in all recorded cells. The best method would be a co-expression of fluorescent CDKL5 and mAChR proteins via IRES sites in the plasmids and only patch from those cells that 1) have Ca/Ba currents, 2) show fluorescent for both other proteins. Otherwise, co-expression should be shown at least with specific antibodies, and the limitations should be discussed.

The same applies to the next paragraph and experiments with D2 receptors.

Line 183: Delete ',is`' at the end of this line.

For the gain-of-function results without CDKL5- and mAChR-mediated phosphorylation, the relative smaller size of the effects compared to those of the genetic variants in CACNA1E should be mentioned, but I completely agree, that the effects are similar in principle.

Discussion and before: The ',left shift`' of the activation curve should be designated as a ',hyperpolarizing shift`'.

Reviewer #2 (Remarks to the Author):

In this manuscript, the authors identified the voltage-gated Ca²⁺ channel Cav2.3 (encoded by CACNA1E) as a novel physiological target of CDKL5 in mice and humans by using phosphoproteomics and other approaches. The authors further performed recombinant channel electrophysiology and interdisciplinary characterization of Cav2.3-phosphorylated mice and showed that loss of Cav2.3 phosphorylation leads to acquisition of channel function through slow

inactivation and enhanced cholinergic stimulation, resulting in increased neuronal excitability. The characteristics of unphosphorylated Cav2.3 are very similar to those described for CACNA1E gain-of-function mutations that cause DEE69, a disease that shares clinical features with CDD. Throughout this study, the necessary experiments have been covered and the conclusions are well supported by the data obtained. If the following points are appropriately corrected, it is considered to be worthy of publication.

- (1) The data obtained for Fig. 1a are so clear as to be difficult to obtain in this type of experiment, and strongly support the conclusion that the physiological substrate of CDKL5 is Cav2.3. However, it is necessary to show that the protein expression level of Cav2.3 is not altered by CDKL5-KO.
- (2) Regarding the method section, there are some parts where the description is insufficient or necessary papers are not cited. For example, methods to accurately generate SILAC ratios using stable isotope-labeled amino acids as light-labeled AAs, even in non-dialysed serum or with inadequate labeling efficiency, have been published and should be cited. (i) Nat Methods, doi: 10.1038/nmeth0907-677.(ii) Mol. Biosyst.,DOI: 10.1039/B921379A.
- (3) Please cite the paper also for the Phosphopeptide enrichment protocol.
- (4) Insufficient MS conditions, e.g., AGC and CE values, m/z scan range, etc.
- (5) No description of LC for LC/MS. Please include the instrument name, flow rate, gradient, mobile phase, injection volume, etc.
- (6) Database for database search should be described not only for mouse but also for human. Not only the name, but also its version and number of sequences.
- (7) For the raw data PRIDE repository: identify the URL and access key for reviewers.

Reviewer #3 (Remarks to the Author):

CDKL5 deficiency disorder (CDD) is a severe X-linked neurological condition affecting primarily young female patients, caused by mutation in the cyclin-dependent kinase-like 5 (CDKL5) gene. Only a few physiological substrates of CDKL5 are currently known, which hampers the discovery of therapeutic strategies for CDD. Here, Sampedro-Castañeda and colleagues presented the identification of a novel target of CDKL5, the voltage-gated Ca²⁺ channel Cav2.3, and argued that inactivation and enhanced cholinergic modulation of unphosphorylated Cav2.3 may contribute to CDD symptoms. The paper is well written, and the authors should be commended for including all methodological details. However, while the electrophysiological studies are convincing and support the conclusions drawn by the authors, the characterization of the Cav2.3 S15A mouse model is not informative, and does not allow us to conclude that Cav2.3 gain-of-function underlies CDD clinical features. The manuscript needs conclusive experiments and could benefit from additional data interpretation.

Major points

- 1- Gender-based differences in rodent behavior are well known. Recent experiments have shown differences in the way male and female rodents respond to stress. For instance, after classical fear conditioning, female mice appear more cautious than males in new situations. Researchers have also discovered sex differences in decision-making and spatial navigation as well as social and running behaviors. Therefore, it is not advisable to combine the behavioral results obtained from different genders, even because the estrous cycle greatly influences behavior. If I understand correctly, the behavioral studies were done on WT and homozygous S15A Cav2.3 mice of both sexes, but it is not clear whether the data presented in Fig. 4 and Supplementary Fig. 4 derived from results obtained in males or females, or combined genders; moreover, it is not specified how many animals per single gender were used. I recommend showing behavior outcomes for all behavioral tests separately for each gender.
- 2- I suggest using a power analysis to estimate the smallest sample size needed for the behavioral experiments. Probably a minimum number of 6-8 animals per experimental group (including gender) are needed.

3- When comparing the Cav2.3 phosphorylation levels of Cdkl5 KO mice (Fig. 1d) and S15A Cav2.3 mice (Fig. 3a), the Cav2.3 phosphorylation in heterozygosity seems more similar to what occurs in the absence of CDKL5. It would be useful to compare the brain levels of pS15 Cav2.3, run on the same western blot, of extracts from Cdkl5 KO mice, and Cav2.3 S15A phosphomutant mice of the same age. A behavioral study of S15A Cav2.3 heterozygous mice compared to Cdkl5 KO mice seems much more appropriate and informative to define the role of Cav2.3 gain-of-function in CDD pathology.

4- How do authors explain the presence of a specific band corresponding to the phosphorylated form of Cav2.3 in the homozygous mouse model (Fig. 3a,c), and the different height (migration) on gel for the total Cav2.3 protein compared to the S15A Cav2.3 phosphorylated one? It appears that the two protein bands do not match in relation to the molecular weight marker (Fig. 1d, Fig. 3a).

5- As stated before, it is now not possible to conclude that loss of pS15 Cav2.3 is a key contributor to the phenotypic features of CDD due to the very few behavioral similarities between the two mouse models. Cdkl5-null mice have a much more severe phenotype, as published in several papers that were not cited in the manuscript. Therefore, the conclusion that there are striking behavioral similarities between Cav2.3 phosphomutant mice and Cdkl5 KO mice is not assertable. Of note, cognitive impairment, assessable with learning and memory tests such as the Y- and Barnes mazes, was not present in Cav2.3 phosphomutant mice (Supplementary Fig. 3), differently from what was shown in the CDD mouse model.

Although patients with CACNA1E mutations have a partially overlapping clinical phenotype with CDD, at present only evidence of the effect of Cav2.3 inhibitors in reversing the pathological phenotype in vitro or in vivo CDD models would confirm the involvement of Cav2.3 gain-of-function in CDD.

6- The sentence in the discussion: "The difference between males and females, observed in both ECoG and behavioural seizure analysis, may arise from increased basal function of Cav2.3 in the female brain downstream of hormonal signals." must be justified with a reference in this regard. Can it be ruled out that the gender diversity may depend on the small number of animals used?

7- The authors are correct, the identification of key physiological targets of CDKL5 directly involved in the regulation of cellular excitability may help elucidate the epileptogenic mechanisms underlying the CDD phenotype. Cav2.3 could be a good candidate underlying neuronal hyperexcitability in the absence of CDKL5. Adult Cav2.3 S15A mice, as reported for Cdkl5 KO mice, did not show spontaneous seizures, making this correlation difficult. However, the recent finding that aged heterozygous Cdkl5 mutant mice exhibit spontaneous epileptic spasms might be informative in this regard. Are there indications regarding Cav2.3 expression and S15 phosphorylation levels in relation to mouse age? Do aged Cav2.3 phosphomutant mice exhibit spontaneous seizures? A more in-depth investigation and discussion of this aspect could improve the manuscript.

Minor points

1- How was the sequence of behavioral tests organized to minimize the possibility of one test influencing the subsequent evaluation of the next test? Please add this information to the Methods section.

2- Cdkl5 KO mice are differently indicated (Cdkl5 KO, CDKL5 KO) across the text. Please homologue, I would suggest using Cdkl5 KO mice.

3- The number of samples or animals in the figure and supplementary fig. legends were indicated with both n= and N=. Please homologue, I would suggest using n =.

4- Two-Way ANOVA or TW ANOVA in the figure and supplementary fig. legends.

Point-by-point response to Reviewers' comments**REVIEWER COMMENTS****Reviewer #1 (Remarks to the Author):**

This review was done by Holger Lerche from Tübingen University, Germany.

The authors present comprehensive evidence in experiments ranging from molecular to both mouse and human cellular and whole animal levels, that Cav2.3 is a target of CDKL5 and that CDKL5-related DEE is at least partially caused by the lack of CDKL5-related phosphorylation of Cav2.3 and also a functionally related enhanced cholinergic stimulation of non-phosphorylated Cav2.3 channels. It is an impressive amount of work adding important novel information to the field. The manuscript is very clearly written with nice and clear figures illustrating the results. The methodology is sound except some minor issues. I have only few comments which are outlined as follows.

We would like to thank the reviewer for their positive and constructive comments. We have addressed their comments in our responses and with added material to the manuscript. These changes have strengthened and improved our paper and we hope they will be satisfactory for the reviewer.

Comments

Line 52: ...aetiology OF approximately...

This has been amended in the manuscript.

Line 88: Ref 44 does not provide functional studies on Cav2.3 channels, so should be cited only for confirming genetic findings in CACNA1E (as in line 87 and later).

Agreed. This has been amended in the manuscript.

The overlap between phenotypes of patients with CACNA1E- and CDKL5-related DEE, and also the differences, should be given in more detail. Maybe a supplemental table could be provided to give an easily accessible overview of both similarities and differences of both diseases. This should be also part of the discussion.

As suggested, we have prepared a new supplementary Table 1 summarizing the most common clinical features of CDKL5 and CACNA1E neurodevelopmental disorders, as reported to date. We hope that this will highlight differences and similarities between both disorders, although it is important to emphasize that there is significantly less literature regarding CACNA1E encephalopathy when compared to CDKL5 deficiency disorder and hence the table is only accurate as far as is possible to infer from the relatively limited number of reported cases harbouring CACNA1E variants. The manuscript has been amended in the introduction to reflect addition of Supplementary Table 1. "Patients with CACNA1E mutations have overlapping clinical phenotypes with CDD, such as intractable seizures, profound intellectual disability and hypotonia (new Supplementary Table 1)."

The discussion's first paragraph is edited to include further information: "These include a high incidence of intractable seizures, global developmental delay, intellectual disability, autistic behaviours, a range of motor phenotypes such as hypotonia, hyperkinetic movements and

stereotypies, as well as sleep, visual and sensory disturbances. The cellular and circuit basis of these clinical manifestations remains to be dissected in both neurodevelopmental disorders. CDKL5 deficiency is multifactorial as many kinase targets exhibit reduced phosphorylation. Nevertheless, a consistent feature of the Cav2.3 variants associated with epilepsy studied to date is GoF of the channel, by shifting the activation voltage to more hyperpolarized potentials, altering maximal current and/or by slowing inactivation (Helbig KL et al 2018).”

It should be explicitly mentioned in the text, that S14 and S15 are corresponding serine residues in human and murine Cav2.3 channels.

We have clarified this point in the text upon first mention of the identified phosphorylation site in the Results section.

“Strikingly, the murine Cav2.3 S15 site (corresponding to S14 in humans) matches the RPXS/T* consensus motif...”

Lines 167 and following paragraph: The authors should discuss and include additional evidence, that all three proteins (Cav2.3, CDKL5, mACHR) are expressed in their transfected HEK cells. I appreciate that there was an additional measurable functional effect of mACHR-related phosphorylation (hyperpolarizing shift of the activation curve), but it is not clear if all three proteins were really expressed in all recorded cells. The best method would be a co-expression of fluorescent CDKL5 and mACHR proteins via IRES sites in the plasmids and only patch from those cells that 1) have Ca/Ba currents, 2) show fluorescent for both other proteins. Otherwise, co-expression should be shown at least with specific antibodies, and the limitations should be discussed.

The same applies to the next paragraph and experiments with D2 receptors.

We would like to thank the reviewer for highlighting this point. We agree with the necessity to co-transfect our plasmids into HEK293 cells and confirm co-expression. We adopted the following measures during our experiments:

1) We obtained stable cell lines (Tetracycline-inducible HEK293 cells expressing human Cav β 3 and α 2 δ 1 subunits & HEK293 cells stably expressing human Cav b1b and α 2 δ 1 subunits) to ensure that all channels would have similar α 1E+ α 2 δ 1 subunit composition across experiments and to reduce the number of plasmids to transfect.

2) We transfected GFP at low levels when compared to CDKL5 and CACNA1E (α 1E-HA, FLAG-CDKL5 full length and GFP (pcDNA3) were co-transfected at a ratio of 10:4:1 (HEK β 3 cell line) or 8:4:1 (HEKb1b cell line), to ensure that when GFP positive cells are selected for electrophysiology, there is a high probability of plasmid uptake for the most abundant constructs (Cav2.3, CDKL5) In other words, if a cell has successfully incorporated and expressed low-abundance GFP (as far as can be comfortably detected under the patch microscope), it is highly likely to have also incorporated the other, more abundant plasmids.

3) We would like to note that in the absence of CACNA1E there would be no Calcium/ Barium channel currents recorded, virtually all GFP+ cells we recorded from had Calcium/Barium currents.

We used immunostainings to confirm co-expression levels in our multi-plasmid transfection experiments. As requested by the reviewer, we have now included representative immunocytochemistry images obtained from two independent transfections (identical to our GPCR electrophysiology datasets (Figure 2g-p and supplementary Figure 2g-j). Our results demonstrate that

all GFP-positive cells also express Cav2.3, CDKL5 and M3 or D2 receptors (new Supplementary Figure 3).

In the new Supplementary Figure 3a, quadruple transfection of GFP, HA-Cav2.3, CDKL5 and M3 receptor are shown. All GFP+ cells express the other 3 plasmids. Side panels (2b) show background staining in the absence of transfection. Note the presence of endogenous M3 receptor expression, as reported (Rumenapp et al 2001). In new Supplementary Figure 3c, we demonstrate a triple transfection of D2 receptor tagged with GFP (GFP-D2), CDKL5 and HA-Cav2.3. All GFP+ cells also express Cav2.3 and CDKL5 indicating a high level of transfection.

These results underscore the efficacy of the transfection strategy described above. Protocols have been added to Methods and limitations discussed in Discussion paragraph 4, in the revised manuscript.

4) As noted by the reviewer, although M1/M3 receptor is not fluorescently tagged, protein expression and integrity can be clearly distinguished in our recordings both by Cav2.3 current enhancement and by the hyperpolarizing shift in the IV curve upon carbachol application. Similarly, D2 receptors, which were GFP tagged, produce a reliable and reversible change in current amplitude following quinirole application, allowing us to verify their correct expression and function. As for CDKL5, expression and integrity cannot be verified on a cell by cell basis, and the possibility remains that there are varying degrees of expression of this protein in our recordings, although the data as a whole shows clear differences between the groups.

Line 183: Delete ,is' at the end of this line.

Thank you. This has been amended.

For the gain-of-function results without CDKL5- and mAChR-mediated phosphorylation, the relative smaller size of the effects compared to those of the genetic variants in CACNA1E should be mentioned, but I completely agree, that the effects are similar in principle.

This has been changed in the text.

Discussion and before: The ,left shift' of the activation curve should be designated as a ,hyperpolarizing shift'.

This has been changed in the text.

Reviewer #2 (Remarks to the Author):

In this manuscript, the authors identified the voltage-gated Ca²⁺ channel Cav2.3 (encoded by CACNA1E) as a novel physiological target of CDKL5 in mice and humans by using phosphoproteomics and other approaches. The authors further performed recombinant channel electrophysiology and interdisciplinary characterization of Cav2.3-phosphorylated mice and showed that loss of Cav2.3 phosphorylation leads to acquisition of channel function through slow inactivation and enhanced cholinergic stimulation, resulting in increased neuronal excitability. The characteristics of unphosphorylated Cav2.3 are very similar to those described for CACNA1E gain-of-function mutations that cause DEE69, a disease that shares clinical features with CDD.

Throughout this study, the necessary experiments have been covered and the conclusions are well supported by the data obtained. If the following points are appropriately corrected, it is considered to be worthy of publication.

We thank the reviewer for their positive review. We have addressed their specific comments with additional information included in materials and methods as well as in the text.

(1) The data obtained for Fig. 1a are so clear as to be difficult to obtain in this type of experiment, and strongly support the conclusion that the physiological substrate of CDKL5 is Cav2.3. However, it is necessary to show that the protein expression level of Cav2.3 is not altered by CDKL5-KO.

We would like to thank the reviewer for their insightful review of our paper and kind words about the quality of the results. We fully agree that it is very important to show that Cav2.3 protein expression is not altered upon the loss of CDKL5. Otherwise, the decrease in Cav2.3 phosphorylation could potentially also be attributed to the decrease in total protein level. The design of the SILAC experiment leading us into the direction of Cav2.3 as a substrate of CDKL5 unfortunately did not allow us to measure total protein levels in parallel, as the phosphopeptide-enriched samples obtained in this experiment were unsuitable for accurate measurement of total protein levels. However, the even distribution and below significance values of 6 different Cav2.3 phosphorylation sites (see Reviewer figure 1) made us confident that specifically Cav2.3 S15 phosphorylation is affected by *Cdkl5* knockout (KO), independent of changes in total protein level. We understand that each candidate from a mass spectrometry screen needs to be validated, with HEK293, mouse and human neuron data (new Figure 1 c,d,e), we validated Cav2.3 as a bona fide CDKL5 substrate. In addition, we extensively studied Cav2.3 S15 phosphorylation and total protein levels using specific antibodies in *Cdkl5* KO mice and CDD patient-derived cell lines. We did not observe changes in Cav2.3 protein levels in any of these experiments either (new Figure 1d and new supplementary Figure 4).

Reviewer figure 1: Representation of the results in figure 1A highlighting all identified Cav2.3 (Cacna1e) phosphorylation sites. Only S15 is significantly downregulated, while other sites are not significantly changed. These findings indicate that Cav2.3 protein level is not altered upon loss of CDKL5.

(2) Regarding the method section, there are some parts where the description is insufficient or necessary papers are not cited. For example, methods to accurately generate SILAC ratios using stable isotope-labeled amino acids as light-labeled AAs, even in non-dialysed serum or with inadequate labeling efficiency, have been published and should be cited. (i) Nat Methods, doi: 10.1038/nmeth0907-677.(ii) Mol. Biosyst.,DOI: 10.1039/ B921379A.

We are sorry to hear that the reviewer believes our method section was insufficiently describing the SILAC experiment. We addressed these concerns in responses to reviewers' comments 3-7 to the best of our ability. We acknowledged the concern about labelling efficiency and Arginine to Proline

conversion raised here and would like to stress that extensive quality control has been performed on these samples before mixing. For all samples >90% labeling efficiency and <10% Arginine to Proline conversion was observed. This information is now included in the methods section titled "SILAC phosphoproteomics sample preparation".

We have now added two more citations for the methods we used to achieve this quality, these are included in the same methods section:

"...Based on methods to achieve accurate SILAC ratios in non-dialyzed serum (van Hoof et al., 2007, Imami et al., 2010), cells were grown in neurobasal media free of L-Arginine and L-Lysine (Invitrogen) supplemented with either L-Lysine-8 (U-13C6, U-15N2) and L-Arginine-10 (U-13C6, U-15N4) or L-Lysine-4 (4, 4, 5, 5-D4) and L-Arginine-6 (U-13C6)(Cambridge Isotope Laboratories). L-Proline (200mg/ml) was supplemented to prevent Arginine to Proline conversion. Each sample was checked for adequate SILAC label incorporation (>90%) and minimal Arginine to Proline conversion (<10%)..."

(3) Please cite the paper also for the Phosphopeptide enrichment protocol.

Phosphopeptide enrichment was performed according to a protocol first used in Swaffer et al., Cell (2016). We included this reference and more details about the sample preparation in general in the methods section:

"Samples were prepared for mass spectrometry as previously described in Swaffer et al., 2016. In brief, protein lysates were mixed in a 1:1 ratio to obtain 3x heavy WT/light KO and 3x heavy KO/light WT samples containing 1 mg total protein. Mixed samples were reduced with 5 mM DTT, alkylated with 10 mM iodoacetamide and quenched with 7.5 M DTT prior to o/n digestion trypsin at 37C. Peptides were desalted using 50 mg Sep-Pak C18 Cartridges (Waters) and vacuum dried completely before further digestion at 37C using Lys-C for 2 hrs, followed by o/n trypsin digestion. Peptides were desalted and dried again. Mixing and digestion checks were performed after each digestion step. Samples were enriched for phosphopeptides according to an in-house protocol using 5 mg of Titansphere titanium dioxide beads (GL Sciences) incubated for 10 min with peptides resuspended in loading buffer (80% acetonitrile, 5% trifluoroacetic acid, 1M glycolic acid). TiO2 beads were subsequently washed with loading buffer, wash buffer 1 (80% ACN 1% TFA) and wash buffer 2 (10% acetonitrile, 0.2% trifluoroacetic acid), and peptides were finally eluted using 1% ammonium hydroxide and 5% ammonium hydroxide subsequently. Combined phosphopeptide eluates were vacuum dried completely and desalted using C18 Stage Tips and vacuum dried completely. Samples were stored at -80C until required for analysis by mass spectrometry."

(4) Insufficient MS conditions, e.g., AGC and CE values, m/z scan range, etc.

(5) No description of LC for LC/MS. Please include the instrument name, flow rate, gradient, mobile phase, injection volume, etc.

We have included the requested information in the method section. More specific details can also be found in the raw files available in the PRIDE repository.

"Samples were analysed by online nanoflow LC-MS/MS using an LTQ Orbitrap Velos mass spectrometer (Thermo Scientific) coupled to an Ultimate 3000 RSLCnano (Thermo Scientific). Resolubilised sample (10 µL per injection in 1% aqueous trifluoroacetic acid, TFA) was loaded via autosampler into a 20 µL sample loop and pre-concentrated onto a cartridge trap column 300 µm I.D. x 5 mm, packed with Acclaim PepMap100 C18, 5 µm, 100Å using the loading buffer, 2% v/v acetonitrile, 0.05% v/v trifluoroacetic acid, 97.95% water (Optima grade, Fisher Scientific) at a flow rate of 20 µL/min for 2 min in the column oven held at 40 °C. Peptides were backflushed and gradient eluted onto a PepMap

RSLC C18 75 μm x 50 cm, 2 μm particle size, 100 \AA pore size, reversed phase EASY-Spray analytical column (Thermo Scientific) at a flow rate of 250 nL/min and with the column temperature held at 40 $^{\circ}\text{C}$, and a spray voltage of 2 kV using the EASY-Spray Source (Thermo Scientific). Gradient elution buffers were A 0.1% v/v Formic Acid, 5% v/v DMSO, 94.9% v/v water and B 0.1% v/v Formic Acid, 5% v/v DMSO, 20% v/v water, 74.9% v/v acetonitrile (all Optima grade, Fisher Scientific aside from DMSO, Honeywell Research Chemicals). The gradient elution profile was 2% B to 30% B over 143 minutes, then 30% B to 50% B over a further 20 minutes. The mass spectrometer was operated in data dependent acquisition mode with the top 10 most abundant peptides selected for MS/MS by either collision-induced dissociation, multi-stage activation or higher-energy collisional dissociation fragmentation techniques. The three instrument methods used a capillary temperature of 275 $^{\circ}\text{C}$, an MS1 Orbitrap scan resolution of 60,000 FWHM at m/z 400, mass range 300-2000 m/z , S-Lens RF level 60%, FTMS Full AGC target $1e6$, maximum injection time 500 ms and spectra were acquired in profile. Only precursors with charge state >2 were permitted for selection for fragmentation and dynamic exclusion was enabled to exclude after $n=1$ times within 20 s for 20 s. Fragmentation was subsequently performed on all selected precursor masses and the MS2 scan data was acquired in centroid mode. For the method with CID activation MS2 scans were acquired in the linear ion trap following CID fragmentation with normalized collision energy of 30%, Ion Trap MSn AGC target $1e4$ and maximum injection time 100 ms. For the method with CID activation and multi-stage activation enabled parameters were as for CID but with normalized collision energy of 35% and a neutral loss mass list (32.70, 49, 98). For the method with HCD activation MS2 scans were acquired in the Orbitrap at a resolution of 7500 FWHM at m/z 400, following HCD fragmentation with a normalized collision energy of 45%. The parameters used for the HCD MS2 scan were fixed first mass 100 m/z , FTMS MSn AGC target $5e4$, maximum injection time 500 ms.”

(6) Database for database search should be described not only for mouse but also for human. Not only the name, but also its version and number of sequences.

The phosphoproteomics SILAC experiment was performed using mouse primary neurons. Therefore, the database search was only performed against a mouse database and no human database was used. We have included the version and number of sequences of the used mouse database:

“...the database search was performed using the Andromeda search engine against the Mus Musculus canonical sequences downloaded from UniProtKB (August 2012, 77938 sequences).”

(7) For the raw data PRIDE repository: identify the URL and access key for reviewers.

We sincerely apologise to the reviewers for not providing access to the PRIDE repository immediately. These details had been lost somewhere in the submission process. We completely acknowledge the importance of access to raw data, especially in the case of large omics datasets, and believe you have received the URL and access key by now. We also included the dataset identifier and relevant references in the method section:

“The mass spectrometry proteomics data have been deposited to the ProteomeXchange Consortium via the PRIDE (Perez-Riverol et al., 2022) partner repository with the dataset identifier PXD038505.”

Reviewer #3 (Remarks to the Author):

CDKL5 deficiency disorder (CDD) is a severe X-linked neurological condition affecting primarily young female patients, caused by mutation in the cyclin-dependent kinase-like 5 (CDKL5) gene. Only a few

physiological substrates of CDKL5 are currently known, which hampers the discovery of therapeutic strategies for CDD. Here, Sampedro-Castañeda and colleagues presented the identification of a novel target of CDKL5, the voltage-gated Ca²⁺ channel Cav2.3, and argued that inactivation and enhanced cholinergic modulation of unphosphorylated Cav2.3 may contribute to CDD symptoms. The paper is well written, and the authors should be commended for including all methodological details. However, while the electrophysiological studies are convincing and support the conclusions drawn by the authors, the characterization of the Cav2.3 S15A mouse model is not informative, and does not allow us to conclude that Cav2.3 gain-of-function underlies CDD clinical features. The manuscript needs conclusive experiments and could benefit from additional data interpretation.

We are grateful to the reviewer for their time and constructive feedback on our manuscript. We have carefully addressed the comments with new data, analysis and discussions. We believe that in our revised manuscript we present a comprehensive analysis of the Cav2.3 phosphomutant mouse model with conclusive experiments.

Major points

1- Gender-based differences in rodent behavior are well known. Recent experiments have shown differences in the way male and female rodents respond to stress. For instance, after classical fear conditioning, female mice appear more cautious than males in new situations. Researchers have also discovered sex differences in decision-making and spatial navigation as well as social and running behaviors. Therefore, it is not advisable to combine the behavioral results obtained from different genders, even because the estrous cycle greatly influences behavior. If I understand correctly, the behavioral studies were done on WT and homozygous S15A Cav2.3 mice of both sexes, but it is not clear whether the data presented in Fig. 4 and Supplementary Fig. 4 derived from results obtained in males or females, or combined genders; moreover, it is not specified how many animals per single gender were used. I recommend showing behavior outcomes for all behavioral tests separately for each gender.

We would like to thank the reviewer for their insightful comments highlighting important considerations.

When designing our experiments, we did not have any reason to expect CACNA1E S15A mice to have sex-dependent differences in behavioral tests. We noted the recommendations on the importance to include animals of both sexes in rodent behavioural studies. Thus, we created mixed sex cohorts that were relatively balanced (*). Combined sex groups analysis was presented in the first version of our manuscript and we understand that this should have been reflected and detailed in original figure legends.

We agree with the reviewer that sex can be a source of variation. This is relevant in the context of CDD, where phenotypic differences between genders have emerged in mouse models, although there is a bias for more data being obtained from males. Our original number of animals was not sufficient to analyse potential behavioural differences between males and females due to low numbers per sex. We have now addressed this gap in our study by performing new behaviour experiments with additional mice. We have analyzed sex differences separately in behaviour datasets (presented in the new Figure 4 and new supplementary Figure 6). Some behaviour tests show differences due to sex x genotype (sociability, home cage locomotion, Barnes maze), and sometimes the variation is due exclusively to genotype (fear conditioning, wheel use). As previously observed, some behaviours are not affected at all in the phosphomutant model. We have opted to present all data separated by sex

as suggested by the reviewer. In addition, for fear conditioning analysis we also grouped the sexes as both sexes showed a similar phenotype. The new numbers of animals/sex/genotype (**) are now stated clearly in legends. Our results with larger mouse numbers allow us to reach conclusions on behavioural phenotypes.

*Numbers of mice used in behaviour tests in our original manuscript:

males WT=3, HOM=3; females WT=5, HOM=4

** Number of mice in our revised manuscript:

Males WT=6, HOM=5; females WT=8, HOM=6

We use littermate wild type and S15A HOM mice; experimental animals are obtained by crossing Cav2.3 S15A HET mice. This was a limiting factor in the number of mice we could obtain, when compared to keeping separate colonies for wild type and mutant mice.

Home-cage cohort: males WT=5, HOM=4; females WT=4, HOM=5, are unchanged between the first and revised versions. We were unable to add more animals to the existing DVC dataset due to equipment unavailability, but we have presented the existing data separated by sex groups with corresponding numbers and statistics for transparency (new Figure 4 and new Supplementary Figure 6).

2- I suggest using a power analysis to estimate the smallest sample size needed for the behavioral experiments. Probably a minimum number of 6-8 animals per experimental group (including gender) are needed.

In order to estimate the minimum number of animals required to detect differences due to genotype and/or sex, we have performed a post-hoc or retrospective power analysis using the G*Power analysis tool (University of Dusseldorf, (Faul *et al*, 2007)(now cited in our new manuscript in methods section) and the observed effect size from our original ANOVA test (fear conditioning dataset). This analysis, done exclusively to help design prospective experiments with a comparative level of power, revealed that increasing our experimental group to 6 animals/gender/genotype would yield an acceptably powered dataset (76% chance of detecting an effect) assuming the same effect size of 8% as measured when the genders were grouped together. We then carried out a comprehensive, blinded and unbiased battery of behavioural tests with a second cohort of WT and homozygous phosphomutant S15A Cav2.3 mice to add to our previous data. We looked carefully at the outputs of the Two- and Three-Way ANOVA for this new dataset to check for possible interactions between all factors. These results have been detailed in the new figure legends. The clear differences found with our new analysis suggest that our data is now sufficiently powered to detect sex x genotype effects.

3- When comparing the Cav2.3 phosphorylation levels of Cdkl5 KO mice (Fig. 1d) and S15A Cav2.3 mice (Fig. 3a), the Cav2.3 phosphorylation in heterozygosity seems more similar to what occurs in the absence of CDKL5. It would be useful to compare the brain levels of pS15 Cav2.3, run on the same western blot, of extracts from Cdkl5 KO mice, and Cav2.3 S15A phosphomutant mice of the same age. A behavioral study of S15A Cav2.3 heterozygous mice compared to Cdkl5 KO mice seems much more appropriate and informative to define the role of Cav2.3 gain-of-function in CDD pathology.

We thank the reviewer for their careful and important comment. We agree that there is a similarity between pS15 levels in *Cdkl5* KOs (original Figure 1d,g) and heterozygous phosphomutants (original Figure 3a-c). We would caution that the lysates used for these *in vivo* validations are not directly

comparable, because they were derived from P20 cortex (*Cdkl5* KOs) or 5-6-week-old whole brain hemispheres (phosphomutants).

In order to clearly address the levels of pS15 in these two mouse models, we have now added new experiments. We have performed Western blot analysis of age-matched cortical lysates from both *Cdkl5* KO and S15A phosphomutant mouse models (new Supplementary Fig. 4). Our results show that, at 5-6 weeks-old S15A heterozygous mice have approximately 50% of pS15, which would be expected as one of the two copies of CACNA1E can be phosphorylated. We observed that *Cdkl5* KOs have ~15% of pS15 remaining. The levels of pSer15 in S15A heterozygous mice are significantly higher than *Cdkl5* KO mice. Phospho S15 Cav levels reach background levels (<10%) in pS15 homozygous mice, as expected. Our new data clarifies the amount of pCav2.3 phosphorylation that remains in *Cdkl5* KO mice. In light of these results we postulate that Cav2.3 homozygous phosphomutant mice are a suitable model to study the role of CDKL5 mediated phosphorylation on Cav2.3 pathophysiology.

Additionally, it should be noted that we chose to study HOM S15A mice in order to get a clear picture of the physiological relevance of this phosphorylation in mice without confounding effects of heterozygosity.

4- How do authors explain the presence of a specific band corresponding to the phosphorylated form of Cav2.3 in the homozygous mouse model (Fig. 3a,c), and the different height (migration) on gel for the total Cav2.3 protein compared to the S15A Cav2.3 phosphorylated one? It appears that the two protein bands do not match in relation to the molecular weight marker (Fig. 1d, Fig. 3a).

We thank the reviewer for their careful review and pointing out this important point.

In the first version of our manuscript, the Cav2.3 total antibody we used was a side-product antibody obtained during our custom-made pSer15 phospho-specific antibody production. The protocol for generation of phosphospecific antibodies involves multiple steps. After immunization of the rabbits with pSer15 peptide, the serum is subjected to the first column purification step using a non-phosphorylated form of the Cav2.3 N-terminal peptide. Antibodies that bind the column are eliminated from the pSer15 antibody pool as they would not be phosphospecific. These "side-product" antibodies can in principle be used as a total Cav2.3 antibody, as they are recognizing the N terminus of Cav2.3. This antibody is what we used in the initial version of our manuscript. However, as the reviewer carefully pointed out, the band that is recognized by our custom Cav2.3 total antibody (essentially Ser15 non-phospho antibody) seems to be of a higher molecular weight than that detected by the phosphospecific antibody. While we don't know the exact reason for this observation, we speculate that the non-phosphorylated form of Cav2.3 could have other modifications that cause it to have a different molecular weight. For the sake of this manuscript, we have now removed all data obtained with the custom-made Cav2.3 total antibody and instead used a commercial Cav2.3 antibody as a total Cav2.3 control.

We have now carried out new experiments using a commercial antibody against mouse Cav2.3 (Synaptic Systems 152411) presented in the new Figure 1 d,g, new Figure 3 a-c (& also new Supplementary Figure 4 and new Supplementary Figure 7). The commercial Cav2.3 antibody was used for detection of Cav2.3 in human iPSC derived neurons in our original manuscript (Figure 1 e, h). Our new data shows that the pCav2.3 and total Cav2.3 bands are at the same molecular weight, as expected. Our quantifications and conclusions are virtually identical to those presented in the original figure.

5- As stated before, it is now not possible to conclude that loss of pS15 Cav2.3 is a key contributor to the phenotypic features of CDD due to the very few behavioral similarities between the two mouse

models. Cdkl5-null mice have a much more severe phenotype, as published in several papers that were not cited in the manuscript. Therefore, the conclusion that there are striking behavioral similarities between Cav2.3 phosphomutant mice and Cdkl5 KO mice is not assertable. Of note, cognitive impairment, assessable with learning and memory tests such as the Y- and Barnes mazes, was not present in Cav2.3 phosphomutant mice (Supplementary Fig. 3), differently from what was shown in the CDD mouse model.

Although patients with CACNA1E mutations have a partially overlapping clinical phenotype with CDD, at present only evidence of the effect of Cav2.3 inhibitors in reversing the pathological phenotype *in vitro* or *in vivo* CDD models would confirm the involvement of Cav2.3 gain-of-function in CDD.

We thank the reviewer for their valuable comments.

We have now deleted from the 6th paragraph in the Discussion section, the assertion of “We find striking behavioural similarities between our Cav2.3 phosphomutant mice and *Cdkl5* KOs” and replaced it with “We also find some behavioural similarities between our Cav2.3 phosphomutant mice and *Cdkl5* KOs.”

We have included further references that investigated the *Cdkl5* deficient mouse models in behaviour tests to depict the full spectrum of phenotypes in the new version of our manuscript: Awad PN et al 2023, Hao S et al 2021, Mulcahey PJ et al 2020, Viglione A et al 2022, Gao et al 2020, Trazzi et al 2018, Terzic et al 2021, Yennawar et al 2019. Having carried out additional behavioural analysis and increased the numbers of animals in the cohort, we can now conclusively state the specific differences and similarities between Cav2.3 phosphomutant mice and *Cdkl5* KOs. Our new results highlight that spatial (Barnes maze) and fear (contextual fear conditioning) memory, as well as sociability are impaired in both models, suggesting that unphosphorylated Cav2.3 may contribute to these phenotypes. We have specified where differences exist between males and females and summarise our findings in new Figure 4 and new Supplementary Figure 6.

As the reviewer rightly points out, while the evidence we present on the pathological link between CACNA1E/Cav2.3 and CDKL5 deficiency disorder is very compelling, experiments with Cav2.3 inhibitors testing if Cav2.3 inhibition reverses CDD phenotypes in human/mouse models are needed to further confirm this functional link. Unfortunately, there are no Cav2.3 inhibitors with sufficient specificity that would be suitable for *in vitro* or *in vivo* investigations, currently. Existing inhibitors also inhibit other channels, making such experiments ambiguous in interpretation. For example, the most selective and potent inhibitor of Cav2.3, the spider toxin SNX-482, has been shown to be at least 10x more potent on Kv4.3/2 (Kimm & Bean, 2014), an important neuronal voltage gated K⁺ channel mediating the A-current and essential for dendritic excitability. Other non-toxin Cav2 channel inhibitors such as TROX-1 or Physallin F would potentially target presynaptic Cav2.2 and 2.1 (Shan *et al*, 2019) (Swensen *et al*, 2012). A highly specific inhibitor, designed and developed specifically for Cav2.3 is needed to modulate Cav2.3 activity. A comprehensive future study with multiple outcome measures (including behavioural tests) using different doses of a specific Cav2.3 inhibitor, at multiple developmental time points would be needed for deciphering the extent of Cav2.3's contribution to CDD phenotypes in mice. In the new version of our manuscript, in the final paragraph of Discussion section, we included the statement “Specific Cav2.3 inhibitors would be required to test if inhibition of Cav2.3 could ameliorate CDKL5 deficiency phenotypes in mouse and human models.” and clarified the need for Cav2.3 specific inhibitors to confirm the link between CDD and CACNA1E pathology. In addition, it is known that Cav2.3 promotes excitability as Cav2.3 knockout mice are resistant to seizures (Weiergraber *et al*, 2006; Weiergraber *et al*, 2007), therefore a specific effect on Cav2.3

inhibition of CDD would need to be assessed by comparing inhibition of Cav2.3 on control mice and CDD model mice.

6- The sentence in the discussion: “The difference between males and females, observed in both EcoG and behavioural seizure analysis, may arise from increased basal function of Cav2.3 in the female brain downstream of hormonal signals.” Must be justified with a reference in this regard. Can it be ruled out that the gender diversity may depend on the small number of animals used?

We thank the reviewer for their comment. We cannot rule out the possibility that variability and insufficient numbers could have led to the observed difference, although the result seems plausible given our behavioural observations. We have revised this statement to “The difference between males and females, observed in both EcoG and behavioural seizure analysis could reflect biological sex differences related to Cav2.3 function but requires further experiments with larger number of mice to ascertain. Nevertheless, sex-specific differences should be taken into consideration in future *Cdkl5* KO mouse behaviour studies and seizure susceptibility experiments.”

7- The authors are correct, the identification of key physiological targets of CDKL5 directly involved in the regulation of cellular excitability may help elucidate the epileptogenic mechanisms underlying the CDD phenotype. Cav2.3 could be a good candidate underlying neuronal hyperexcitability in the absence of CDKL5. Adult Cav2.3 S15A mice, as reported for *Cdkl5* KO mice, did not show spontaneous seizures, making this correlation difficult. However, the recent finding that aged heterozygous *Cdkl5* mutant mice exhibit spontaneous epileptic spasms might be informative in this regard. Are there indications regarding Cav2.3 expression and S15 phosphorylation levels in relation to mouse age? Do aged Cav2.3 phosphomutant mice exhibit spontaneous seizures? A more in-depth investigation and discussion of this aspect could improve the manuscript.

We thank the reviewer for their comment, we agree that Cav2.3 is a CDKL5 substrate that is a good candidate for explaining CDD patient’s epilepsy phenotypes. Lack of spontaneous seizures in *Cdkl5* KO mice and in Cav2.3 makes this correlation difficult, as stated by the reviewer.

In S15A HOM mice we do not observe cage disturbance-associated seizures up to the age of 40 weeks, as shown by our home-cage monitoring cohort that was kept for an extended period. We have added this information to the manuscript, in Results section.

We are aware of the spontaneous seizures observed in aged heterozygous *Cdkl5* female mice after approximately 20 weeks (Terzic et al 2021 and Mulcahey et al 2020). The changes in Cav2.3 and its phosphorylation levels with age may indeed be an important point to consider.

To address this, we have conducted new experiments and compared Cav2.3 total protein and S15 phosphorylation levels in WT mice at two different adult ages, including an age at which heterozygous *Cdkl5* females exhibit spontaneous seizures. Our data show a significant increase in channel expression in cortex between the ages of P56 (8 weeks) and P154 (22 weeks) (new Supplementary Figure 7). This result agrees with the increasing levels of Cav2.3 protein and CACNA1E mRNA between juvenile (P13) and adult (13 weeks) midbrain dopaminergic neurons (Benkert *et al*, 2019). On the other hand, the levels of pS15 phosphorylation in cortex are reduced at 22-weeks as measured by the ratio of pS15 Cav2.3/total Cav2.3. Whether or not an age-related decrease in the relative Cav2.3 phosphorylation levels contributes to the increased susceptibility to seizures in ageing heterozygous *Cdkl5* female mice (Terzic *et al*, 2021) remains to be investigated. We have included this finding in new Supplementary Figure 7 and we added the following paragraph to the Results section in our new manuscript.

“Finally, since heterozygous *Cdkl5* females have spontaneous seizures starting at approximately 20 weeks and increasing with age (Mulcahey *et al*, 2020; Terzic *et al.*, 2021), we investigated the levels of Cav2.3 phosphorylation in brain by comparing 8- and 22-week-old mice. We found that Cav2.3 total levels are increased and relative levels of pS15 are reduced in 22-week-old mice (new Supplementary Fig. 7), raising the possibility that increased Cav2.3 expression could contribute to late-onset epilepsy in heterozygous *Cdkl5* KO females”.

Minor points

1- How was the sequence of behavioral tests organized to minimize the possibility of one test influencing the subsequent evaluation of the next test? Please add this information to the Methods section.

We have now included the following statement in the Methods section to explain this point in detail:

“Two cohorts of mice were used for all 7 behaviour tests, carried out in the specific order described below, to minimise the possibility of one test influencing the evaluation of the subsequent test. Similarly, to avoid unnecessary stress, mice were allowed to habituate to the testing room for at least an hour before the behaviour tests. Lighting was manipulated depending on the specific behaviour experiment. Aversive light levels were avoided where possible.”

2- *Cdkl5* KO mice are differently indicated (*Cdkl5* KO, *CDKL5* KO) across the text. Please homologate, I would suggest using *Cdkl5* KO mice.

This has been amended.

3- The number of samples or animals in the figure and supplementary fig. legends were indicated with both *n*= and *N*=. Please homologate, I would suggest using *n* =.

This has been amended.

4- Two-Way ANOVA or TW ANOVA in the figure and supplementary fig. legends.

This has been amended.

References

- Benkert J, Hess S, Roy S, Beccano-Kelly D, Wiederspohn N, Duda J, Simons C, Patil K, Gaifullina A, Mannal N *et al* (2019) Cav2.3 channels contribute to dopaminergic neuron loss in a model of Parkinson's disease. *Nat Commun* 10: 5094
- Faul F, Erdfelder E, Lang AG, Buchner A (2007) G*Power 3: a flexible statistical power analysis program for the social, behavioral, and biomedical sciences. *Behav Res Methods* 39: 175-191
- Kimm T, Bean BP (2014) Inhibition of A-type potassium current by the peptide toxin SNX-482. *J Neurosci* 34: 9182-9189
- Mulcahey PJ, Tang S, Takano H, White A, Davila Portillo DR, Kane OM, Marsh ED, Zhou Z, Coulter DA (2020) Aged heterozygous *Cdkl5* mutant mice exhibit spontaneous epileptic spasms. *Exp Neurol* 332: 113388

Shan Z, Cai S, Yu J, Zhang Z, Vallecillo TGM, Serafini MJ, Thomas AM, Pham NYN, Bellampalli SS, Moutal A *et al* (2019) Reversal of Peripheral Neuropathic Pain by the Small-Molecule Natural Product Physalin F via Block of CaV2.3 (R-Type) and CaV2.2 (N-Type) Voltage-Gated Calcium Channels. *ACS Chem Neurosci* 10: 2939-2955

Swensen AM, Herrington J, Bugianesi RM, Dai G, Haedo RJ, Ratliff KS, Smith MM, Warren VA, Arneric SP, Eduljee C *et al* (2012) Characterization of the substituted N-triazole oxindole TROX-1, a small-molecule, state-dependent inhibitor of Ca(V)2 calcium channels. *Mol Pharmacol* 81: 488-497

Terzic B, Cui Y, Edmondson AC, Tang S, Sarmiento N, Zaitseva D, Marsh ED, Coulter DA, Zhou Z (2021) X-linked cellular mosaicism underlies age-dependent occurrence of seizure-like events in mouse models of CDKL5 deficiency disorder. *Neurobiology of disease* 148: 105176

Weiergraber M, Henry M, Krieger A, Kamp M, Radhakrishnan K, Hescheler J, Schneider T (2006) Altered seizure susceptibility in mice lacking the Ca(v)2.3 E-type Ca²⁺ channel. *Epilepsia* 47: 839-850

Weiergraber M, Henry M, Radhakrishnan K, Hescheler J, Schneider T (2007) Hippocampal seizure resistance and reduced neuronal excitotoxicity in mice lacking the Cav2.3 E/R-type voltage-gated calcium channel. *Journal of neurophysiology* 97: 3660-3669

REVIEWERS' COMMENTS

Reviewer #1 (Remarks to the Author):

The authors did a very careful revision addressing all points raised adequately. I have no further comments.

Reviewer #2 (Remarks to the Author):

The authors responded appropriately to this reviewer's comments. When this reviewer accessed the raw MS files in PRIDE, the number of raw MS files (66 files) was different from the 30 files described in the Excel file named "Experimental_Design.xlsx" stored in the same project PXD038505.

Please add the description to the PRIDE excel file.

Reviewer #3 (Remarks to the Author):

I am satisfied with the authors' answers.

Point-by-point response to Reviewers' comments

REVIEWERS' COMMENTS

Reviewer #1 (Remarks to the Author):

The authors did a very careful revision addressing all points raised adequately. I have no further comments.

We thank the reviewer.

Reviewer #2 (Remarks to the Author):

The authors responded appropriately to this reviewer's comments. When this reviewer accessed the raw MS files in PRIDE, the number of raw MS files (66 files) was different from the 30 files described in the Excel file named "Experimental_Design.xlsx" stored in the same project PXD038505. Please add the description to the PRIDE excel file.

We thank the reviewer for their careful comment. The excel sheet in the PRIDE is being modified to correctly reflect the set of files uploaded.

Reviewer #3 (Remarks to the Author):

I am satisfied with the authors' answers.

We thank the reviewer.